# Carbon intensity of global crude oil trading and market policy implications

Yash Dixit [1], Hassan El-Houjeiri[2], Jean-Christophe Monfort [2], Liang Jing [3,4], Yiqi Zhang [5], James Littlefield [4], Wennan Long[6], Christoph Falter [1], Alhassan Badahdah [2], Joule Bergerson [3], Raymond L. Speth [1] ✉ & Steven R. H. Barrett [1]

The energy mix transition has accelerated the need for more accurate emissions reporting throughout the petroleum supply chain. Despite increasing environmental regulations and pressure for emissions disclosure, the low resolution of existing carbon footprint assessment does not account for the complexity of crude oil trading. The lack of source crude traceability has led to poor visibility into the "well-to-refinery-entrance" carbon intensities at the level of granular pathways between producers and destination markets. Using high-fidelity datasets, optimization algorithms to facilitate supply chain traceability and bottom-up, physics-based emission estimators, we show that the variability in global "well-to-refinery-entrance" carbon intensities at the level of crude trade pathways is significant: 4.2–214.1 kg-$CO_2$-equivalent/barrel with a volume-weighted average of 50.5 kg-$CO_2$-equivalent/barrel. Coupled with oil supply forecasts under 1.5 °C scenarios up to 2050, this variability translates to additional $CO_2$-equivalent savings of 1.5–6.1 Gigatons that could be realized solely by prioritizing low-carbon supply chain pathways without other capital-intensive mitigation measures.

Petroleum fuels account for 32% of the global primary energy supply[1]. Future projections under 1.5 °C climate scenarios range from plateauing to decreasing supply, corresponding to up to 40% reduction from present levels[1–4]. Moreover, certain applications, such as aviation and petrochemicals, have limited short-term, scalable alternatives. On this rapidly-changing market backdrop, crude extraction and transportation together account for ~1.9 Gigatons/year (Gt/year) of $CO_2$-equivalent ($CO_2$eq), based on the International Energy Agency's (IEA) recent assessment[5]. These emissions are distributed across a complex global trade network—hundreds of crude blends are transported over thousands of pipeline miles and millions of ocean shipping miles via complex interconnected trade networks and refined at hundreds of refineries[6–10]. Thus, carbon intensity (CI) based differentiation through

better emissions estimation and reporting at the level of supply chain pathways can be an effective strategy to optimize the remaining use of crude oil and thus contribute to a low-carbon future.

Although regulatory efforts such as the Low Carbon Fuel Standard (LCFS) by the California Air Resources Board (CARB), the Fuel Quality Directive by European regulators, and the CORSIA (Carbon Offsetting and Reduction Scheme for International Aviation) criteria have sought to use this differentiation as a policy tool, they have been hindered by data gaps and inadequate supply chain traceability[11–14]. Furthermore, while existing literature is directionally congruent with these policy goals, it is limited by methodological challenges. Studies based on the Oil Production Greenhouse Gas Emissions Estimator (OPGEE) model[15], an open-source bottom-

[1]Laboratory for Aviation and The Environment, Department of Aeronautics and Astronautics, MIT, Cambridge, MA, USA. [2]Energy Traceability Technology, Technology Strategy and Planning Department, Aramco, Dhahran, Saudi Arabia. [3]Department of Chemical and Petroleum Engineering, University of Calgary, Calgary, AB, Canada. [4]Climate and Sustainability Group, Aramco Research Center–Detroit, Aramco Americas, Novi, MI, USA. [5]Division of Environment and Sustainability, The Hong Kong University of Science and Technology, Hong Kong, China. [6]Energy Science & Engineering, Stanford University, Stanford, CA, USA. ✉e-mail: speth@mit.edu

up emission estimator for upstream oil and gas operations, have illustrated the heterogeneity of crude production CI at the level of individual oil fields[16,17]. However, these emission estimates have not been translated to the level of marketable crude oil blends, thus curtailing the design of policies seeking to incentivize demand for low-carbon crude oil. Additionally, these studies have used fixed and approximated baseline values for crude oil transportation emissions based on models such as the Greenhouse Gases, Regulated Emissions, and Energy Use in Technologies model (GREET)[15,18].

Consequently, these studies have focused more on quantifying opportunities for emissions reductions in the oil and gas sector as a whole and less on introducing incentives for low-carbon practices in the global crude oil trade. The latter has the potential to impact the traded price of crude oil in terms of differentials between different crude grades[3]. The trading market could apply carbon intensity as a specification in crude oil valuation in the same way it considers sulfur content and API density. In such a regulatory environment, crude oil prices would reflect the associated well-to-refinery-entrance CI, with lower CI crudes being traded at a premium to those of higher CI. The blending process is key in establishing traceability—it relates oil fields to blends (e.g., West Texas Intermediate from the U.S. and Ekofisk from Norway), which, together with crude demand at refineries, enables the well-to-refinery-entrance coverage. Thus, a market-based approach measures the CI of crude oil trades from sources (oil fields) to destinations (refineries) by accounting for crude blending and the full impact of crude oil transportation. In summary, existing studies lack

the resolution required for effective policy relevance and exhibit imperfect coverage across the supply chain.

To that end, in this work, we present a global high-resolution assessment of crude oil CI based on bottom-up engineering-based methods. This assessment is based on both public and commercial datasets with optimization algorithms to establish supply chain traceability. We use bottom-up engineering-based tools, including OPGEE[15]; Crude Oil Pipeline Transportation Emissions Model (COP-TEM), a physics-based estimator for pipeline transport emissions[19]; and a shipping emissions estimator based on AIS-tracking data[20,21].

## Results and discussion

### Scope and resolution of the life cycle assessment (LCA)

This analysis uses life cycle assessment (LCA) to account for the well-to-refinery-entrance greenhouse gas (GHG) emissions from the petroleum supply chain. The foundation of the LCA is a network representing the global oil supply chain where oil fields, shipping terminals, pipeline stations and refineries are nodes; pipelines and shipping routes are edges. At the level of each producing country, the network is jointly used with a multi-objective optimization algorithm to estimate crude blending. In conjunction with data on crude demand at refineries, this enables a resolution at the level of individual supply chain pathways, as illustrated in Fig. 1.

The "well-to-refinery-entrance" scope points toward two emission categories—crude extraction (upstream) and crude transportation (midstream). The crude blending algorithm, in conjunction with

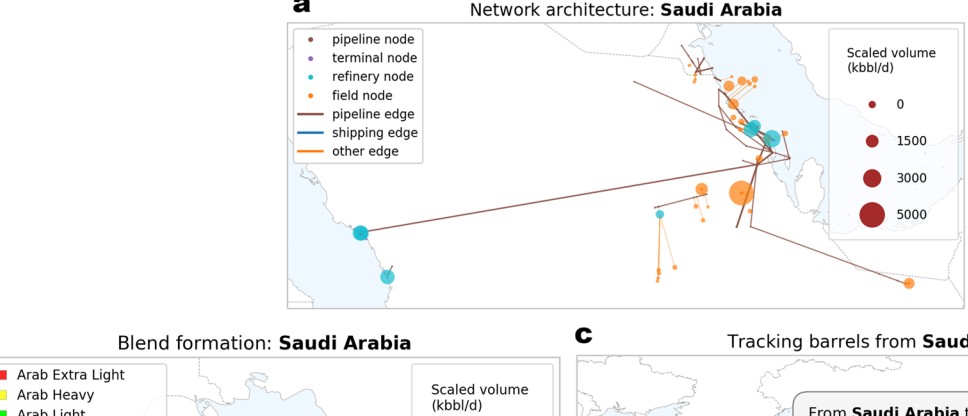

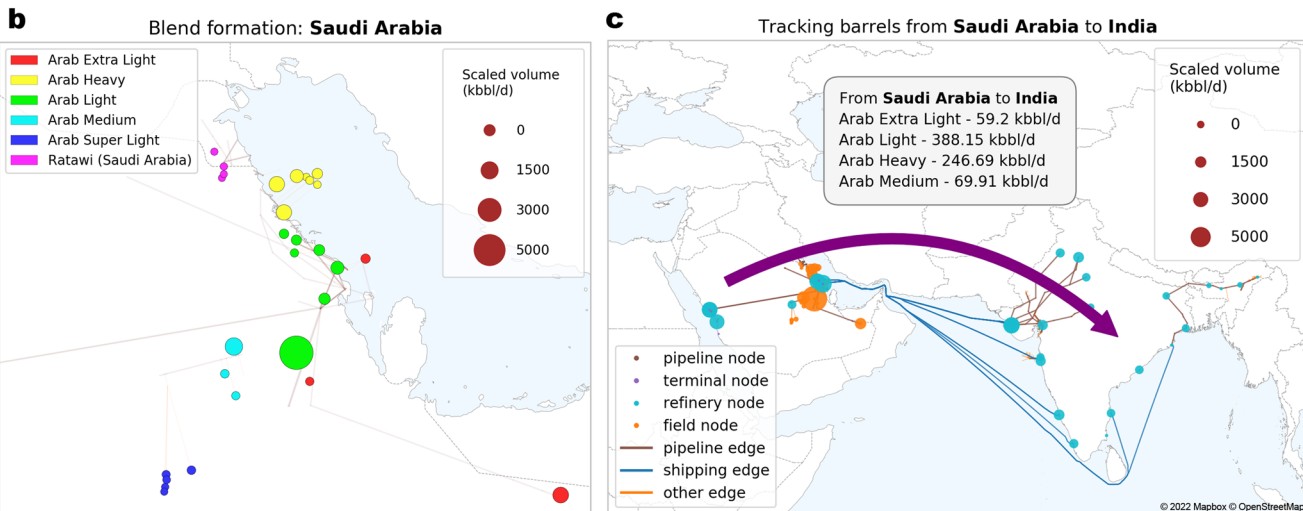

**Fig. 1 | Case study of crudes from Saudi Arabia to India: the supply chain network, blend formation and tracking of crude barrels from sources to destinations (volume shown in kilo-barrels/day or kbbl/d).** Our method captures the global supply with a well-to-refinery-entrance scope. To detail its different aspects, the infrastructure details for Saudi Arabia and its connectivity to India are shown here for illustrative purposes. As shown by the network architecture (**a**), the supply chain consists of nodes (oil fields, shipping terminals, pipeline stations and refineries) and edges (pipelines and shipping routes). The sizes of field nodes and refinery nodes are scaled in proportion of production volume and total intake volume, respectively. To estimate blend formation, we use the network with a multi-objective optimization algorithm that estimates how crude from oil fields combines to form blends (**b**). Tying these elements with information about crude demand at refineries, we track crude at the level of individual supply chain pathways from fields to refineries (**c**).

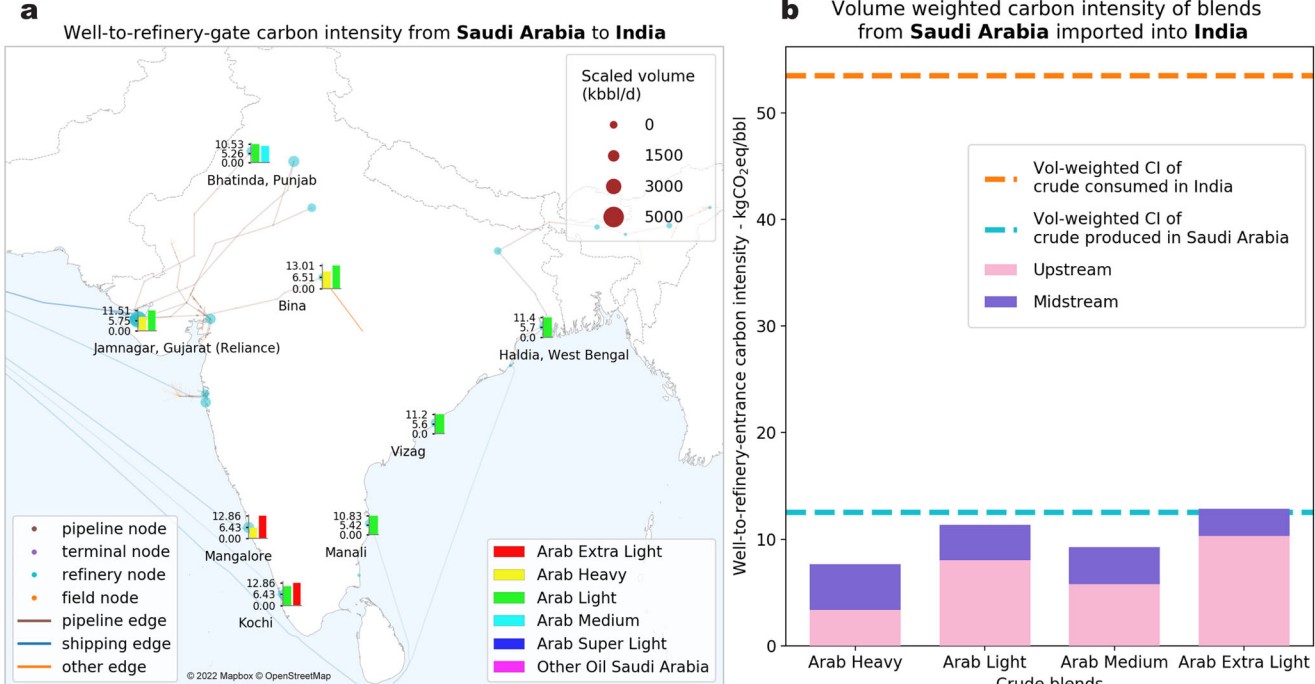

**Fig. 2 | Case study of crudes from Saudi Arabia to India: carbon intensity (CI) estimates (in kg-CO₂eq/barrel) at different levels of policy-relevant aggregation (sample refineries in India selected based on refining volume >20 kilobarrels/day or kbbl/d).** After estimating the upstream and midstream carbon intensities for every individual pathway, we aggregate them at the refineries (**a**); the bars shown at the chosen refinery nodes represent the blend-level well-to-refinery-entrance CI in kg-CO₂eq/bbl. The weighted carbon intensities of crude blends from Saudi Arabia to India (**b**) are not only variable across the relevant four crude blends but also exhibit a wider variability when accounting for different refinery destinations. This demonstrates that the heterogeneity in upstream and midstream emissions leads to each refinery in a country having a unique profile of crude blend carbon intensities.

field-level crude extraction CI calculated by OPGEE[16], constitutes the former, while the latter is estimated using mode-specific emission models (refer to "Methods"). The resolution of the LCA enables the accounting of life cycle emissions at different levels of aggregation. As shown in Fig. 2, this not only generates estimates of carbon intensity for global crude blends but also destination-specific CI to inform policymaking addressing refineries and/or petroleum products.

**Carbon intensity of marketed global crude oil blends**

Figure 3 shows the volume and upstream CI of marketable global crude blends, where the latter is computed by coupling the blend estimation algorithm with the field-level crude production CI. Uncertainties are quantified by varying parameters of the algorithm weighting different factors, such as proximity, pipeline connectivity, etc., in the optimization approach (refer to Supplementary Note 4). We examine blend-level variability within and between countries in addition to the aggregated country-level variability.

Blends in Russia show less inter-blend variation in carbon intensities—standard deviation of 3.32 kg-CO₂-equivalent/barrel (kg-CO₂eq/bbl) versus the global standard deviation of 32.08 kg-CO₂eq/bbl. In addition, the volume-weighted country upstream CI of 48.38 kg-CO₂eq/bbl is close to the global volume-weighted average of 45.03 kg-CO₂eq/bbl. This is due to the low standard deviation in the field-level CI (volume-weighted country standard deviation of 12.76 kg-CO₂eq/bbl versus the global standard deviation of 34.49 kg-CO₂eq/bbl) and the presence of proximate blend clusters connected by common large-scale infrastructure such as the ESPO pipeline network. More generally, the former is the key driver behind inter-blend variability and uncertainties. For example, the low inter-blend variability in Angola can be contrasted to the high inter-blend variability in Canada based on the respective volume-weighted field-level distributions (The mean and standard deviation of the weighted CI distribution of Angolan fields are

50.34 and 6.88 kg-CO₂eq/bbl, respectively, whereas for Canadian fields are 71.73 and 46.62 kg-CO₂eq/bbl, respectively).

With a range of 3.4–181.6 kg-CO₂eq/bbl, the Middle East region shows significant variability, primarily down to differences in field-level CI as described by Masnadi et al.[16] The uncertainties in the CI of Iranian blends are, in general, higher than some of the other major producing countries (major countries defined as the top 15 oil-producing countries as indicated in Fig. 3) due to the greater number of blends (~2.5 million-barrels/day spread over 11 blends) in the country (and less degree of differentiation between crude properties, which makes the blending algorithm sensitive to the weighting parameters. On the other hand, the presence of a predominant blend in Saudi Arabia (~9.8 million-barrels/day spread over 6 blends) and Iraq (~3.1 million-barrels/day spread over 5 blends), namely Arab Light and Basrah Light, respectively, results in low uncertainties. More generally, the presence of fewer blends and one predominant blend indicates lower uncertainties due to the resulting stability of optimal solutions found through the gradient-descent approach (refer to "Methods").

A similar degree of inter-blend variability is seen across Latin America—blends from Mexico, Brazil and Argentina are found to be near the global volume-weighted average, whereas Venezuelan blends have significantly higher CI due to the heavy oil type of reservoirs and the use of carbon-intensive operational practices (e.g., steam flooding)[16].

In North America, the energy-intensive Oil Sands Synthetic blend from Canada has the highest carbon intensity among the major global blends (144.5 kg-CO₂eq/bbl). This closely tracks the fields with similar API density from the oil sands region, which shows a carbon intensity range of 82.5–160.2 kg-CO₂eq/bbl and a volume-weighted mean of 139.3 kg-CO₂eq/bbl, thus attesting to the efficacy of the blending algorithm.

## Upstream carbon intensity and volume of crude blends

Blend-specific aggregation of CO₂eq emissions associated with crude oil extraction

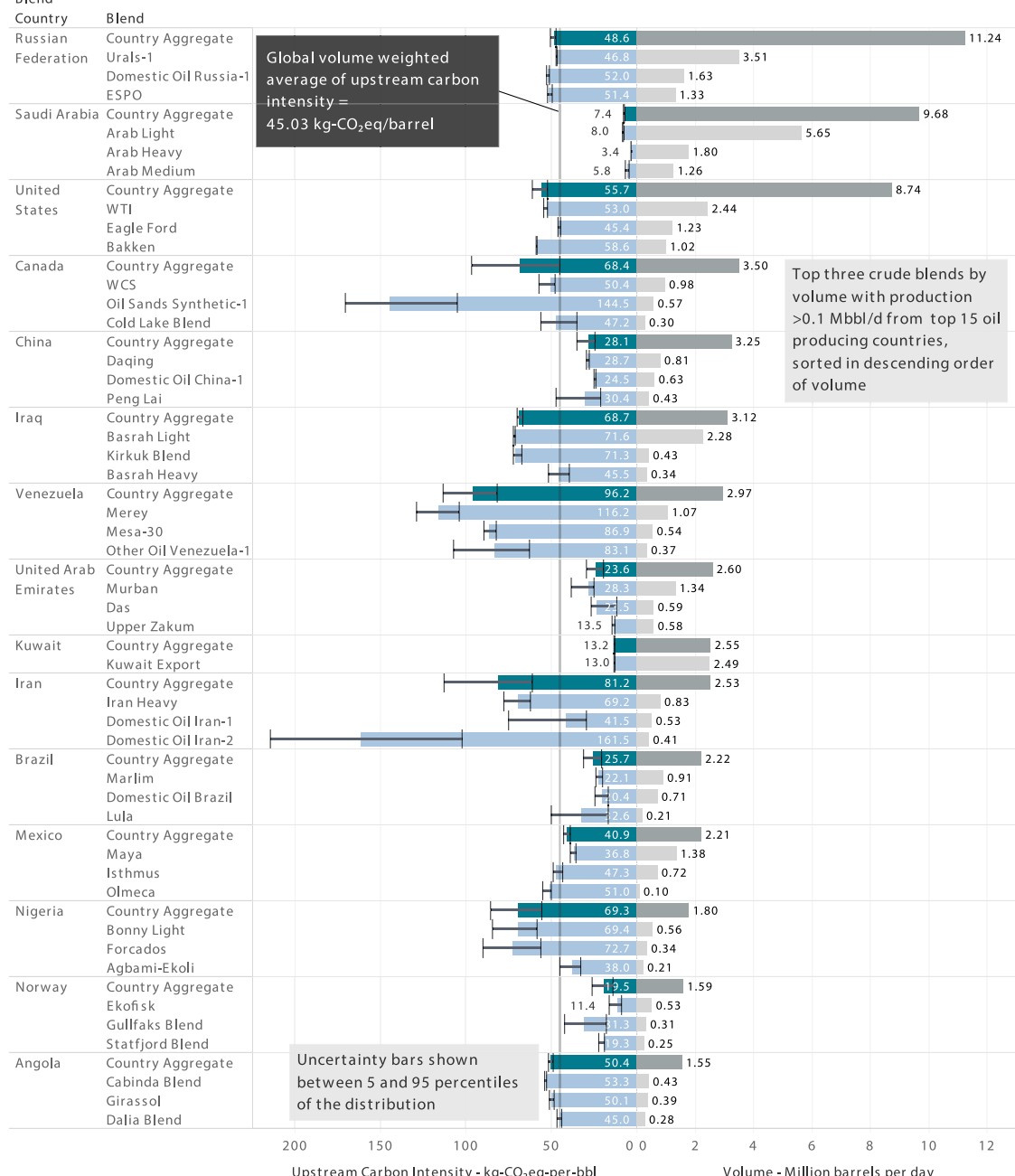

**Fig. 3 | Carbon Intensity (CI) associated with crude extraction aggregated at the level of crude blends (top 3 blends by volume with >0.1 million barrels/day or Mbbl/d) in the top 15 oil-producing countries.** The key blends (based on the aforementioned criterion), their CI, the aggregate country-level volumes and CI are shown for the chosen set of countries. This illustrates how producing countries compare against the global average and how blends compare against the respective country averages. Note that some countries (Russian Federation and Angola) exhibit low uncertainty in carbon intensity for each crude blend and low variability across crude blends; conversely, other countries (Canada, Venezuela, and Iran) have wide uncertainty within and wide variability across crude blends. Source data are provided as a Source Data file.

Figure 4 shows the cumulative well-to-refinery-entrance CI at the blend level by combining the upstream and midstream CI for the 20 highest volume global crude blends, in addition to showing the variability within midstream emissions. The global volume-weighted midstream CI of 5.37 kg-CO₂eq/bbl, as shown in the sub-figure, contributes ~10% to the well-to-refinery-entrance emissions (refer to Supplementary Note 8 for comparisons with relevant literature). Although in magnitude, the average upstream CI is 9 times the midstream CI, the variation in midstream CI, for a given blend, across all supply chain pathways is significant, as shown in the right sub-figure. All the distributions in the chosen set are asymmetrical and have long tails indicative of the complexity of crude transportation networks; these skewed, irregular patterns emphasize the need to identify specific opportunities for policy intervention instead of applying a blanket approach.

Notable examples showing high variability include West Texas Intermediate (WTI) from the U.S. and Maya from Mexico. Given that

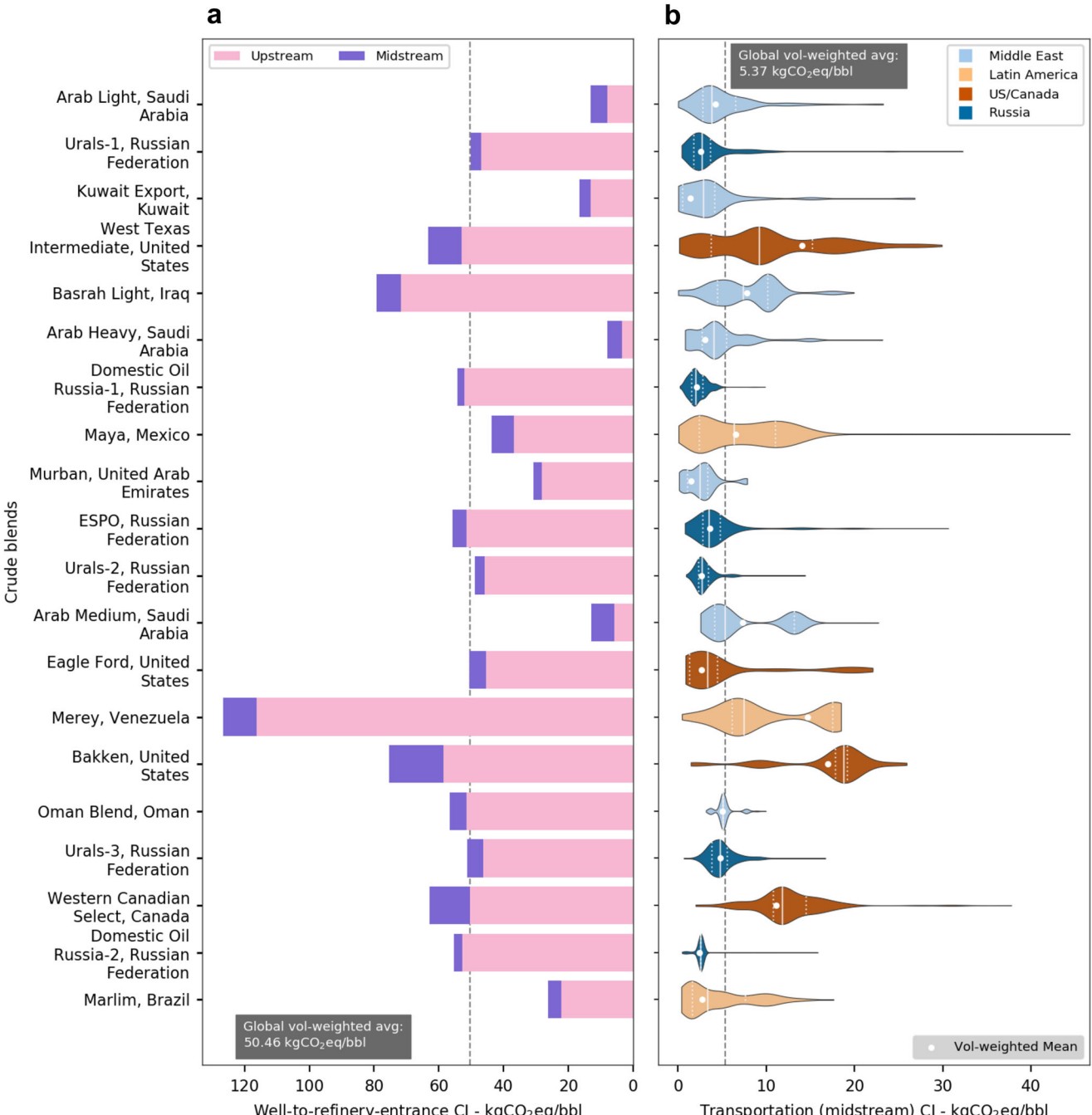

**Fig. 4 | Well-to-refinery-entrance carbon intensity (CI) with the variability in crude transportation CI (in kg·CO₂eq/barrel) for the top 20 global crude blends by volume.** Segmenting well-to-refinery-entrance carbon intensity into upstream and midstream (**a**) demonstrates the wide variability in CI, with Arab Heavy and Western Canadian Select representing the low and high bounds, respectively. This sub-figure also illustrates how the proportion of upstream and midstream CI varies across blends. Violin plots (**b**) show the distribution of midstream CI. Specifically, they illustrate the volume-weighted distribution of midstream CI values (thicker parts of the violin indicate higher probabilities) that the listed crude blends exhibit across different supply chain pathways in the global network. The dashed black line shows the global volume-weighted average, the white dots show the blend-specific volume-weighted averages, and the dashed white lines show the volume-weighted quartiles for each blend. Source data are provided as a Source Data file.

WTI is a benchmark blend centralized in Cushing, Oklahoma, and that it is consumed in 49 North American refineries, the corresponding midstream entails high variability in pipeline miles traversed through an extensive and well-connected pipeline transport system. While for Maya, the variation is explained by a large spread of destinations ranging from domestic refineries to shipped exports to Southeast Asia. Like WTI, the Bakken blend shows high transportation CI due to the long distances between the source fields (Bakken region in Central

North America) and destination refineries, which are as far out as the Gulf Coast and the East Coast of the U.S.

Comparing the midstream CI distributions, we find that blends with a large export footprint, e.g., Arab Medium (100% exported), Merey (>93% exported), and Basrah Light (>94% exported), have multi-modal distributions. This is due to the prevalence of shipped exports and specific features of trade lanes connected to the key import hubs across different continents.

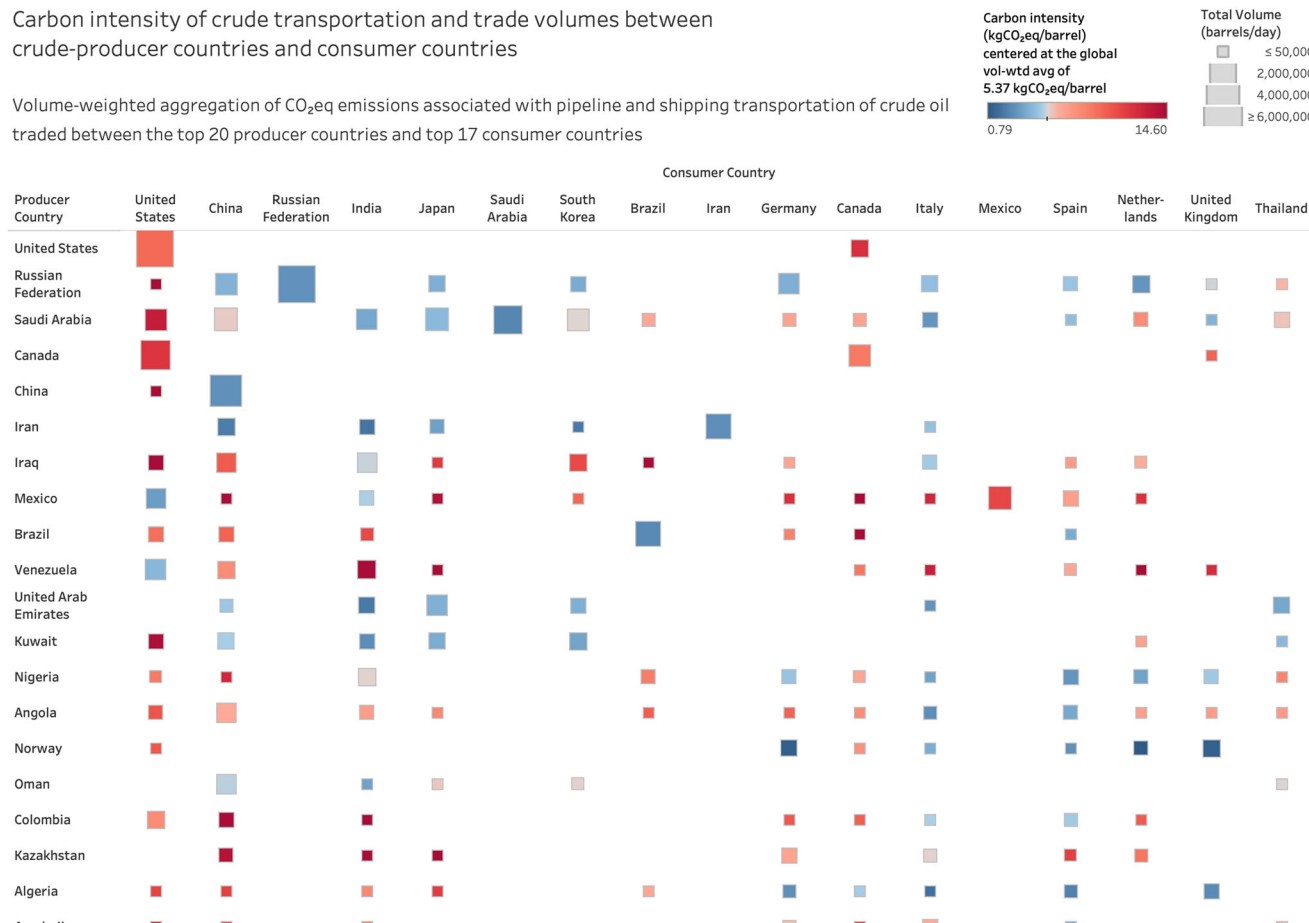

**Fig. 5 | Midstream carbon intensity (CI) and trade volumes between producer and consumer countries.** This figure illustrates how supply chain traceability allows us to see the pairings between producer and consumer countries and the associated trade volumes and carbon intensities for each of these pairings. This is a level of detail aggregated from the individual source blend and destination refinery pairs. Blank values in the visualization matrix correspond to producer, consumer country pairs that do not have a crude trading relationship. Source data are provided as a Source Data file.

## Crude transportation CI from producer to consumer countries

The variability in transportation CI aggregated at the country level shows noticeable patterns in the supply chain (Figs. 5 and 6). Specifically, Fig. 5 illustrates the volume-weighted average CI associated with crude transportation from a given producer country to a given consumer country, while Fig. 6 illustrates the CI trends at the region level.

The extensive pipeline systems in the U.S. and Canada together account for ~40% of the total pipeline miles in the world while representing ~23% of the total refining volume and ~17% of the total crude production volume[22]. These pipeline miles span across a distributed, decentralized network of refineries (~34% of the global number of refineries). In addition, the global volume-weighted average per mile CI of pipeline transport is 2.5 times that of shipping transport. These factors together increase the CI of crude transport in the region to 8.7–12.1 kg-$CO_2$eq/bbl against the global average of 5.37 kg-$CO_2$eq/bbl. In comparison, the CI of the pipeline system in Russia (with extensions into Western/Eastern Europe and China) exhibits a range of 1.5–5.1 kg-$CO_2$eq/bbl, with the differences due to the fact that overall pipeline miles are comparable to crude production (~12% of total global pipeline miles and ~12% of total crude production) as opposed to North America and higher centralization (~62 refineries compared to ~100 in the U.S.). Additionally, given the regions of Eastern Europe, Western Europe and China represent ~88% of Russia's net export volume, the corresponding midstream CI is skewed toward pipeline transport, unlike other exporting regions.

Among shipped exports, as seen in Fig. 6, the volume-weighted shipping CI from Latin America to Asia is 10.7 kg-$CO_2$eq/bbl in contrast to that from the Middle East, which is 5.2 kg-$CO_2$eq/bbl. This difference is attributable to inefficiencies in shipping, the usage of smaller tankers (all things equal, tankers with larger capacities result in lower per-barrel emissions), and longer distances (route carbon intensity has a correlation coefficient of -0.74 with route distance). The differences can also be seen in the country-level breakdown as shown in Fig. 5—for example, CI values from Venezuela to India, Colombia to China, Mexico to Japan are 15.29, 16.05 and 14.10 kg-$CO_2$eq/bbl, respectively; those from Iraq to India, Iran to China, Saudi Arabia to Japan are 5.18, 2.07 and 4.20 kg-$CO_2$eq/bbl, respectively. This is consistent with the patterns in crude tanker activity that indicate high traffic of ultra and very-large crude carriers (ULCCs, VLCCs) with capacities >2 million barrels from the Middle East. Segmenting shipped exports from the Middle East based on destinations, we observe that the CI of trade with North America is two times more than that with South and Southeast Asia. The primary driver behind this difference is the shipping distance—the volume-weighted average shipping distance from the Middle East to North America is 2.5 times the shipping distance to South and Southeast Asia.

Comparing the different sources of transportation emissions (pipeline, shipping and other), we observe that while the global volume-weighted averages for pipeline and shipping are similar (2.55

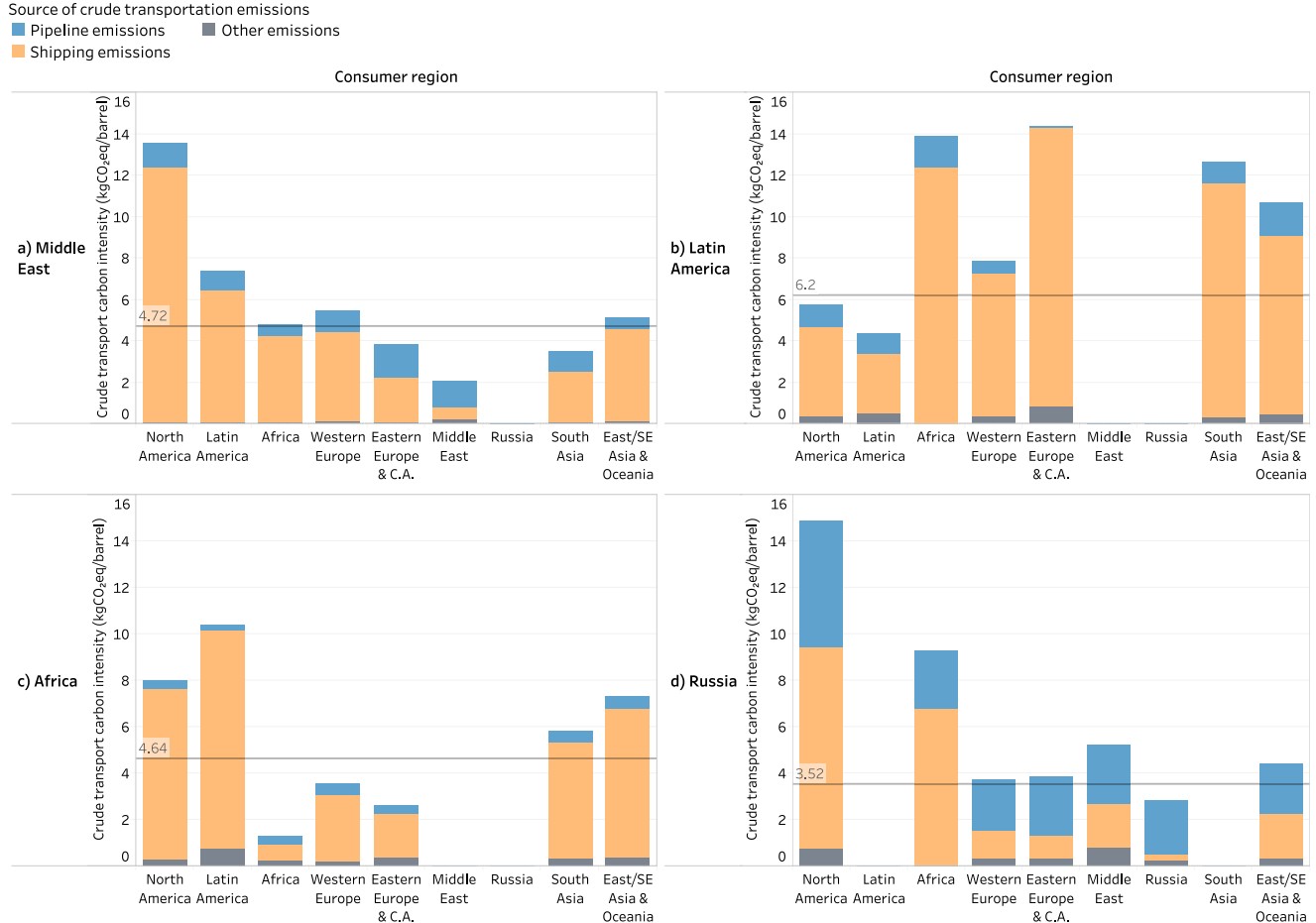

**Fig. 6 | Volume-weighted midstream carbon intensity (CI) in from selected oil-producing regions, segmented by consumer regions and crude transport modes.** Volume-weighted midstream CI in kg-$CO_2$-equivalent/barrel (kg-$CO_2$eq/bbl) from the Middle East (**a**), Latin America (**b**), Africa (**c**) and Russia (**d**) segmented by consumer regions and crude transport modes. Midstream characteristics are highly variable, making the life cycle carbon intensities attributed to crude transportation highly dependent on the consumer region. The main drivers guiding this heterogeneity are total shipping distances, the proportions of pipeline and ocean transport and the overall transport efficiency. (C.A.−Central Asia, SE Asia−Southeast Asia) Source data are provided as a Source Data file.

and 2.61 kg-$CO_2$eq/bbl), there exist significant inter-regional differences as discussed above and demonstrated by Fig. 6. On a global basis, the shipping emissions estimated here are within the bounds of previous studies, where these emissions have been estimated to be up to 14% higher or 20% lower than the present results (refer to Supplementary Note 8 for the detailed comparison of these CI values with relevant literature). Note that the "other" category represents the heuristic edges in the network, which can be conceptually interpreted as the intra-field pipeline connections and approximations to substitute for missing pipeline data. The global volume-weighted average for the "other" edges in the network is 0.21 kg-$CO_2$eq/bbl, i.e., ~8% of the core pipeline emissions (refer to "Methods" and Supplementary Note 3 for more details about the network creation process).

These averages can be further contextualized through the scale of the different modes of transportation in terms of barrel·miles per day: ~33,000 for pipeline, ~145,000 for shipping (and 3000 for other) and CI per unit distance: 5.56 kg-$CO_2$eq/bbl per thousand miles for pipeline (hence also for the "other" category, refer to Supplementary Note 1) and 2.28 kg-$CO_2$eq/bbl per thousand miles for shipping.

**Net carbon footprint attributed to consumer countries**
Upon aggregating the well-to-refinery-entrance CI at all global refineries, Fig. 7 illustrates the attribution of $CO_2$eq emissions to countries based on the crude blends they consume. The map shows all countries

with refining capacity included in the assessment. We further examine countries that process >1 million barrels of crude per day (which cumulatively represent ~78% of the total refining volume included in the study) in the accompanying bar chart and sort them in order of the corresponding net annual kg-$CO_2$eq emissions. Compared to the global volume-weighted average of 50.46 kg-$CO_2$eq/bbl, among these countries, the well-to-refinery-gate carbon intensity varies from 8.84 to 86.39 kg-$CO_2$eq/bbl.

We observe that among producers, the net well-to-refinery-entrance carbon footprint aggregated at the country level based on crude consumption is a strong function of the domestic upstream CI. The main reasons for this are that, on average, upstream CI makes up for 90% of the net CI as discussed earlier, most major producers consume crude produced domestically, and the presence of established and efficient pipeline routes to key domestic refineries—e.g., Saudi Arabia, Iraq, UAE and Russia all consume crude produced 100% domestically (the U.S. is an exception with a significant import footprint, which thus adds transportation emissions due to the resulting shipping activity). Consequently, countries such as Venezuela, Canada and Algeria, with above-average upstream CI, rank accordingly upon aggregation of the well-to-refinery-gate CI. Furthermore, within this subset, countries that are more expansive (i.e., with relatively greater pipeline miles) have a relatively higher transportation CI, most notably Canada and the U.S. This is based on the

**Fig. 7 | Well-to-refinery-entrance carbon intensity (CI) with the variability in CI and absolute emissions for countries with major refining volume.** The map (**a**) shows the well-to-refinery entrance CI for countries with refinery volumes greater than 20,000 barrel/day. The results in this figure are aggregated at the level of consuming countries and demonstrate wide variability across the world's top crude oil refiners (**b**), ranging from 8.84 to 86.4 kg $CO_2$eq/barrel, among countries with refinery volumes greater than 1 million barrels/day. This variability leads to countries with the lowest CIs having less total emissions attributable to well-to-refinery-gate emissions (**c**) than some other countries with half the refining volume. Source data are provided as a Source Data file.

proportionality between length and emissions observed in the fluid mechanics-based approach used for estimating pipeline emissions (refer to "Methods" and Supplementary Note 7 for more details).

Among countries that are predominantly importers, the variability in the blend upstream CI and the midstream CI associated with specific source-destination routes drives the net CI aggregated at the country level based on the crudes they consume. Additionally, importers relying on shipping have a marginally higher refining-attributed CI, as seen by comparing regions of Western Europe (access to pipeline systems from Scandinavia, Russia) and Asia (sparse pipelines; heavy reliance on shipping for imports).

These findings are relevant to regulators that seek to encourage low-carbon sourcing and supply chain pathway prioritization by differentiating crude blends at the point of refinery intake. While Fig. 7 provides a snapshot of the net annual impact of source crudes, the methodology, resolution of the LCA and the modular nature of the analyses act as enablers for climate-aware trade decisions in the near future.

**Implications for decarbonization policy**

At the highest resolution of the LCA, i.e., individual source field to destination refinery pathways, carbon intensities vary from 0.74 to 39.41 g$CO_2$/MJ with a volume-weighted average of 9.01 g$CO_2$/MJ or 50.46 kg-$CO_2$eq/bbl (this average compares to the IEA's estimate of 57.23 kg-$CO_2$eq/bbl[5]). This heterogeneity in life-cycle emissions represents an untapped decarbonization opportunity that can be realized through policy action.

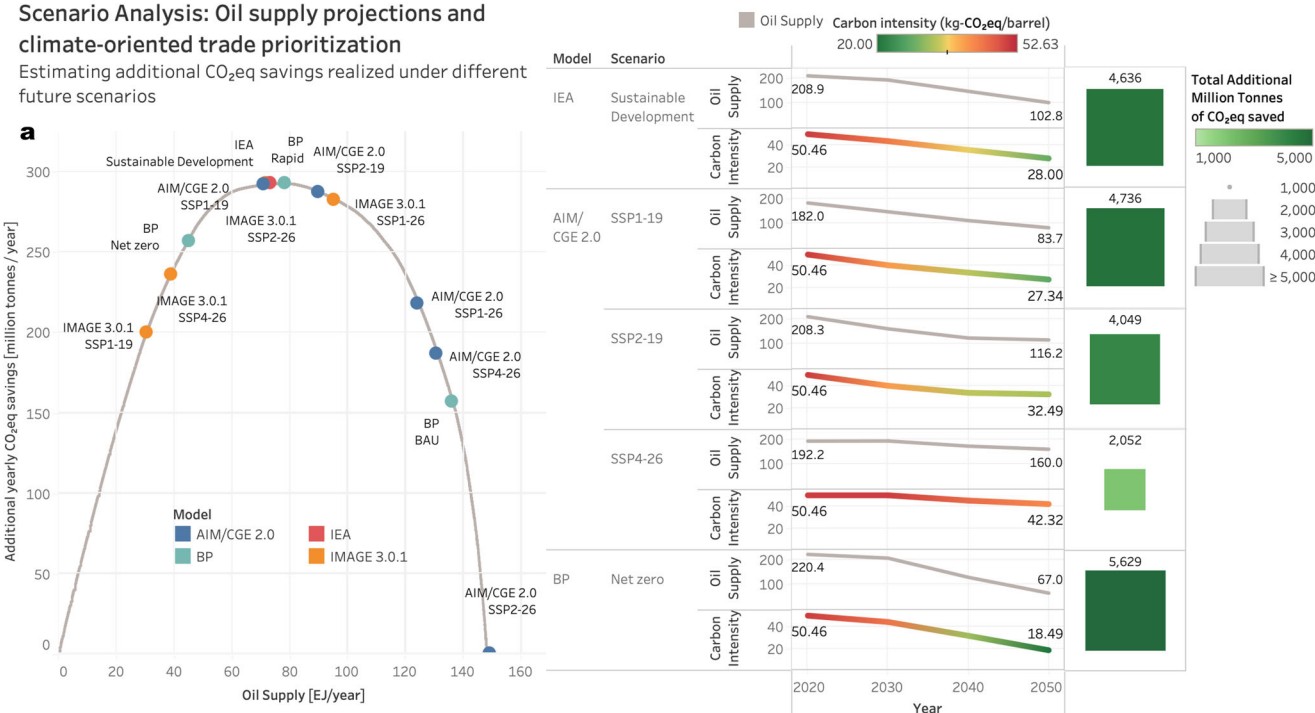

**Fig. 8 | Scenario analysis—oil trade prioritization optimized for life-cycle CO₂eq.** We estimate the total reduction potential in well-to-refinery-gate CO₂eq emissions by analyzing a variety of Shared Socioeconomic Pathway (SSP) scenarios. The estimation is performed by (1) considering the time series of oil supply, (2) ranking crude trade supply chain pathways from highest-to-lowest carbon intensities (CI) and (3) fulfilling supply by prioritizing the low-carbon pathways.

The left sub-figure (**a**) shows this mitigation curve generated by removing marginal barrels of crude at any given supply level based on CI. To plot all scenarios along this curve, we use adjusted volumes, i.e., volumes scaled by the ratio of the scenario-specific 2015 supply value and the net supply from the crude production data to ensure all scenarios have the same starting point. Source data are provided as a Source Data file.

Specifically, through our approach, we fill the gaps related to supply chain traceability which have limited policy efforts such as the LCFS by CARB[11] and the Fuel Quality Directive by European regulators[13] that sought to differentiate between sources of crude. The future success of such policies built on high-fidelity life-cycle assessment could impact crude oil pricing and cause a non-linear shift in the global supply curve. These estimates, along with the high resolution of the LCA, lay the foundation for effective decarbonization policies that can prioritize low-carbon trades. The different levels of aggregation (pathway, blend trade, country) facilitate the flexibility needed to implement these policies. This flexibility could be utilized through multiple policy channels with regional (e.g., CARB in California) and sectoral (e.g., CORSIA by the ICAO for the aviation sector) scopes, thus creating a multi-pronged structure to incentivize CI-based crude differentiation. Furthermore, the significance of these emissions over the 30-year horizon (as shown in Fig. 8) can motivate real-time, granular carbon emissions estimation and reporting from industry. Reporting systems can use appropriate technology (e.g., blockchain)[23], thereby leading to better and more data. This can enable more accurate emission inventories and, in turn, lead to more effective decarbonization through policy action and business strategy. Additionally, with the high granularity and broad coverage of our approach, the analysis creates the framework to fill the gaps that would eventually lead to the convergence of model-based approaches and reporting practices.

To estimate the potential decarbonization impact of the pathway-level CI heterogeneity, we consider oil supply projections from a diverse array of Shared Socioeconomic Pathways (SSP) scenarios[2,24]. We then use the pathway-level CI estimates to meet future demand by prioritizing low-carbon pathways, i.e., fulfilling the reduction in the forecasted annual number of crude barrels by eliminating supply from

pathways having the highest carbon intensities, as illustrated in Fig. 8. The left sub-figure in Fig. 8 shows the different forecasting models and SSP scenarios under consideration with the projected values of oil supply in 2050; we exclusively look at scenarios with future oil supply less than present levels. Next, for every incremental decrease in supply, we quantify the corresponding CO₂eq saved by phasing out the high-carbon supply chain pathways and generate the CO₂eq savings curve shown in the left sub-figure.

For a subset of model/scenario combinations, we then examine the forecasted time series to estimate the average annual carbon intensities with the net CO₂eq savings according to the aforementioned demand fulfillment minimizing overall CO₂eq. Under 1.5 °C scenarios up to 2050, this corresponds to additional CO₂eq savings of 1.5–6.1 Gt with an average of ~4.5 Gt across all models with SSPs 1–4. This is comparable in magnitude to removing ~100 million new gasoline-powered passenger cars, assuming a typical car is driven for 10 years and emits 4.6t CO₂ per year.

It is worth emphasizing that these savings are additional, beyond the savings from reduced supply or CO₂eq emissions management, and thus can be potentially realized without other capital-intensive interventions. Thus, these emissions reduction can be realized in addition to, and not instead of, process-oriented decarbonization options like reduced flaring and carbon capture. Consequently, market-oriented decarbonization based on crude CI differentiation is a valuable piece in the overall decarbonization puzzle (refer to Supplementary Note 9 for additional details on the emissions reduction opportunity).

## Methods
The methods are based on data sources including but not limited to supply chain geo-locations, market trades and shipping routes—commercial data providers include Wood Mackenzie, GlobalData, Kpler,

S&P Global, and public data sources include NASA MODIS and Shuttle Radar Topography Mission (refer to Supplementary Note 1 for more details)[9,10,22,25–27]. They are specifically designed to generate life cycle $CO_2$eq estimates that are actionable from a policy perspective through incentives for improved emissions reporting and best practices. Accordingly, we adopt a methodology that preserves the pathway-level resolution in the supply chain (i.e., granular routes from oil fields to refineries) and is modular in order to have the flexibility of enhancing emission estimates by ingesting augmented data streams (see Supplementary Note 2).

This study uses input data from Masnadi et al.[16] to model the field-level carbon intensities using the latest, most advanced version of the Oil Production Greenhouse Gas Emissions Estimator (OPGEE version 3.0c)[15,28] (See Supplementary Note 10). The upstream CI data generated in this study account for ~98% of the global crude oil and condensate production in 2015, where 95% is crude oil and 3% is condensate. Other data for the modeling of midstream carbon intensities were obtained from commercial data sources for the same year (see Supplementary Note 1). The objective of this work is to understand the physical connections in the crude oil supply chain, making it necessary to allocate production emissions between co-produced oil and condensate. At production, where oil and condensate are in the same stream, we use energy-based co-product allocation to split production emissions between oil and condensate (i.e., the share of emissions assigned to oil are calculated by factoring total emissions by the ratio of energy contained in oil to total oil and condensate energy). This type of allocation is the default method in OPGEE[28]. The oil and condensate streams are routed to different markets after the production stage; all models that we use downstream from production account only for oil flows and do not require us to account for the emissions from condensate or other pre-refinery coproducts. Emissions from natural gas gathering and boosting infrastructure can be a significant contribution to oil and gas production emissions. However, in this study, data are not available for estimating such emissions on a bottom-up basis. OPGEE has a component-level fugitive model that estimates methane emissions from storage tank and equipment leakage based on a comprehensive literature review of component-level activity and emissions measurements[28]. Nonetheless, more data about gathering and boosting infrastructure and fugitive emissions measurements could help improve the accuracy of OPGEE and hence the results of this study (refer to Supplementary Note 11 for more details).

While we perform a static, annualized life cycle analysis, future work needs to account for the impact of temporal variability in the supply chain. Factors such as changing operating conditions at the field level, inventory buffers, dynamic tanker patterns, etc., could have impacts on both traceability and carbon intensity.

Motivated by the complexity and global heterogeneity of crude trade, a network-based approach is used to model the locations, specifications and trade relationships of supply chain assets. Supply chain assets from all aforementioned data sources are consolidated and are assumed to be point geospatial objects. After categorization into the five classes of "fields", "terminals", "shipping ports", "pipeline stations" and "refineries", these objects are encoded as nodes in the supply chain graph. Key node attributes include latitude, longitude, node type (field, pipeline station, shipping terminal, refinery), asset name and asset country. In addition, to facilitate subsequent emission estimation, physical attributes such as annual average ambient temperature and elevation are also included. We then use three classes of edges to create connections between nodes. First, mode edges to represent connections capturing the different modes of transportation (i.e., pipeline and shipping) in the supply chain. Second, concurrent edges to represent connections managing data redundancy and mutual inconsistencies across datasets. Third, heuristic edges for creating rule-based connections to complete the network (e.g., linking fields to proximate pipeline stations). The constructed edges are then given attributes—universal

edge attributes include edge types and distances; type-specific attributes include diameter, length, elevation change for pipeline edges and shipping route distance, and vessel type for shipping edges (refer to Supplementary Note 3 for more details). These edges correspond to the "Other" category mentioned in reporting CI results above.

We exclude barges, trucking and rail transport partly due to the unavailability of data and poor traceability of barge and trucking operations. Moreover, not only do these modes of crude transportation represent a minor share of crude transport compared to pipelines and tankers, but their economic disadvantage (rail transportation cost, on average, is shown to be $5 to $10 per barrel higher than pipeline costs) also makes them merely short-term solutions. For example, in the U.S., where the crude rail network is relatively more mature, rail, barges and trucks combined represent ~10% of the total crude transport based on intake receipts at refineries (the remaining ~90% is through pipelines and tankers)[29,30].

We note that improvements can be made to the supply chain network with respect to data coverage and representation. For example, while oil fields, in reality, are composed of large areas of wells, due to the limited availability of data, we represent them with single geolocation coordinates instead of shapes/areas. Although the impact of the intra-field connectivity to the rest of the network on energy consumption would be low compared to that of the major crude oil pipelines, such spatial modeling can further enhance the fidelity of the network. Furthermore, the availability of data pertaining to granular intra-field pipelines is key in regard to such enhancements.

Next, a "blend estimation" algorithm is designed to predict how crude blends are formed from oil fields by coupling the properties of the global network with a multi-objective optimization approach based on automatic differentiation and unsupervised learning.

Specifically, the blend estimation algorithm is framed as a set of independent country-specific multi-objective optimization problems. For a given oil-producing country, the goal is to estimate the relationship between oil fields and crude blends, which is represented by the configuration matrix Θ that encodes the fraction of volumes from all oil fields contributing to make all crude blends in the given country. **Θ** is a matrix of $F \times B$ dimensions where $F$ is the total number of oil fields and $B$ is the total number of crude blends. The value corresponding to the $i^{th}$ row and $j^{th}$ column is the fraction of crude volume from field $i$ that contributes to blend $j$.

$$\boldsymbol{\Theta} = \begin{bmatrix} \theta_{11} & \cdots & \theta_{1B} \\ \vdots & \ddots & \vdots \\ \theta_{F1} & \cdots & \theta_{FB} \end{bmatrix} \tag{1}$$

The rest of the notation guiding the optimization problem is summarized below, with the dimensions included in parenthesis (see Supplementary Note 4 for more details)

$$\mathbf{V_F} = \text{Volume vector of oil fields}(F \times 1)$$

$$\mathbf{V_B} = \text{Volume vector of crude blends}(B \times 1)$$

$$\mathbf{A_F} = \text{API vector of oil fields}(F \times 1)$$

$$\mathbf{A_B} = \text{API vector of crude blends}(B \times 1)$$

$$\mathbf{D_F} = \text{Distance matrix of oil fields}(F \times F)$$

$$\mathbf{P_F} = \text{Connectivity matrix of oil fields}(F \times F)$$

The goal is to estimate the optimal $\boldsymbol{\Theta}$ i.e., $\boldsymbol{\Theta}^*$ given the following multi-objective cost function containing four sub-costs corresponding to distance ($C_d$), connectivity ($C_c$), volume ($C_v$) and API ($C_a$) (see Supplementary Note 4 for more details).

$$\boldsymbol{\Theta}^* = \text{argmin}_{\Theta}\left[\text{Cost}(C) = \sum_{i \in \{d,c,v,a\}} w_i C_i\right] \quad (2)$$

such that $\forall i, w_i > 0$ and

$$\left[\sum_{i \in \{d,c,v,a\}} w_i\right] = 1; \quad (3)$$

$$C_d = \left|(\boldsymbol{\Theta}\boldsymbol{\Theta}^{\mathbf{T}}) \circ (\mathbf{D_F})\right|_1^1; \quad C_c = \left|(\boldsymbol{\Theta}\boldsymbol{\Theta}^{\mathbf{T}}) \circ (\mathbf{P_F})\right|_1^1 \quad (4)$$

$$C_v = \left\|(\mathbf{V_F^T}\boldsymbol{\Theta})^{\mathbf{T}} - (\mathbf{V_B})\right\|_1^1; \quad C_a = \left\|(\mathbf{A_F^T}\boldsymbol{\Theta})^{\mathbf{T}} - (\mathbf{A_B})\right\|_1^1 \quad (5)$$

where $\circ$ is the Hadamard product and $\|\|_1^1$ is the first norm

The optimization problem is solved using a gradient-based technique coupled with an initialization algorithm. The gradient-based technique uses autodifferentiation with the concept of momentum, which is prominent in the training of deep neural networks[31–33]. The initialization algorithm acts as a bridge between real-world supply chain attributes and the configuration matrix. It ingests information that is not captured by the cost function, such as the similarity between crude blend names and basin and/or oil field names. Furthermore, it includes unsupervised learning and a genetic algorithm to overcome issues of local minima traps encountered in gradient descent.

The output from this algorithm generates estimates of blend-level upstream carbon intensities and high-resolution mapping of crude barrels from sources (oil fields) to destinations (refineries). The latter serves as the input for the barrel tracking algorithm, which finds the shortest paths in the global supply chain network, weighted by pipeline lengths and shipping route distances for pipeline and shipping edges, respectively (see Supplementary Note 5). Lastly, results from the tracking algorithm are used with mode-specific bottom-up models to estimate emissions associated with the transportation of crude oil via pipelines and shipping tankers.

To estimate the GHG emissions associated with crude oil transport via pipelines, we use a first-principles, fluid mechanics-based crude oil pipeline transportation emissions model (COPTEM)[19]. COPTEM is built upon the Energy, Darcy-Weisbach, and Colebrook-White equations, which collectively can be seen as a function of crude oil parameters, pipeline dimensions, and external factors. It is developed to be generalizable for a broad range of crude properties and pipeline dimensions. By default, the model divides any given pipeline into 40 theoretical segments of equal length to mimic a hypothetical distance between two pump stations. Each segment represents 1/40th of the total length of the given pipeline. The energy requirement and GHG emissions associated with maintaining sufficient hydraulic head can then be calculated accordingly. In COPTEM, the energy equation calculates the change in the fluid head between two given points in a pipeline by using friction losses, change in elevation, and change in fluid velocity. The Darcy-Weisbach equation is employed to calculate the friction loss induced by pipeline transportation. As an important factor for the calculation of friction loss, the friction factor is calculated by the recursive Colebrook-White Equation as a function of the pipeline diameter and roughness factor. COPTEM also considers the impact of heat transfer between transported crude and the ambient environment on crude viscosity (see Supplementary Note 7 for more details).

Emissions associated with crude shipping are estimated using a bottom-up estimation based on an integrated dataset of terrestrial and satellite Automatic Identification System (AIS) data along with a global ship parameter database. We process the data by extracting the subset relevant to crude tankers, followed by the categorization of tankers based on size and the identification of trips between shipping terminals. We then model $CO_2$ emissions based on power calculations performed in the processed trip data[20,21]. Specifically, the emissions are estimated as a function of the engine power demand, activity time, and emission factor. The engine power demand for propulsion engines is calculated using the propeller law, which estimates the power associated with propulsion, while the power demand of auxiliary engines and auxiliary boilers is determined according to the corresponding ship class, ship capacity, and activity mode (see Supplementary Note 7 for more details).

The CI estimates of the supply chain should be interpreted separately from the values reported by companies as part of their environmental, social, and corporate governance (ESG) declarations. ESG reporting may have different system boundaries, assumptions, and emissions factors. There are ongoing efforts to close the gap between theoretical models (e.g., OPGEE) and ESG reporting, both by improving LCA models and standardizing ESG reporting practices.

### Reporting summary

Further information on research design is available in the Nature Portfolio Reporting Summary linked to this article.

## Data availability

The upstream CI data for global crude blends generated in this study are provided in Supplementary Note 6. The midstream CI data for global crude blends generated in this study are provided in Supplementary Note 7. The oil field, shipping terminal, pipeline, refinery, crude trade, and shipping route data that support the findings of this study are collectively available from Wood Mackenzie[10], GlobalData[22], Kpler[9], and S&P Global[25]; commercial restrictions apply to the availability of these data, which were used under license for the current study, and so are not publicly available. The earth surface temperature data used in this study are available from NASA MODIS[26]. The augmented elevation data from the NASA Shuttle Radar Topography Mission are available from http://viewfinderpanoramas.org/dem3.html[27]. Source data are provided with this paper.

## Code availability

Analysis was performed in Python using: NetworkX 2.1 for network analysis of the supply chain, PyTorch (torch 1.3.0) for creating the computation graph to perform optimization, Scikit-learn for k-means clustering, and NumPy 1.15.4 and pandas 1.0.3 for data processing. The code is available at https://doi.org/10.5281/zenodo.6814485.

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

## Acknowledgements

This research was supported by a grant from Aramco Americas through the MIT Energy Initiative, which supported Y.D., C.F., R.L.S., and S.R.H.B.; a grant from Aramco Americas to the University of Calgary, which supported L.J. and J.B.; and a grant from Aramco Americas to Stanford University which supported W.L.

## Author contributions

Y.D.: Methodology, Software, Validation, Investigation, Writing—Original Draft; H.E.-H.: Conceptualization, Writing—Review & Editing, Supervision; J.-C. Monfort: Methodology, Data Curation, Writing—Review & Editing; L.J.: Methodology, Software, Investigation; Y.Z.: Methodology, Investigation; J.L.: Investigation, Writing—Review & Editing; W.L.: Methodology, Investigation; C.F.: Methodology, Data Curation; A.B.: Validation, Data Curation; J.B.: Methodology, Supervision; R.L.S.: Conceptualization, Methodology, Writing—Review & Editing, Supervision; S.R.H.B.: Conceptualization, Supervision.

## Competing interests

The authors declare the following competing interests: Y.D., L.J., W.L., C.F., J.B., R.L.S. and S.R.H.B. received funding from grants from Saudi Aramco, which supported this work. H.E.-H., J.-C.M., L.J., J.L., and A.B. are employees of Saudi Aramco or one of its subsidiaries. The remaining authors declare no competing interests.
