## [Peer Review File · Nature Communications]

REVIEWER COMMENTS

Reviewer #1 (Remarks to the Author):

- I think this is a well-done and interesting article that should be published after a few issues are addressed. The analysis is robust and the visuals are excellent. The paper could be improved by building out what the government and corporate implications of this information should be. More specific and actionable recommendations would make the paper more meaningful. Statistics and infographics are great, but what should I do with them?

- While the LCA techniques seem robust, the authors should address how the values might be different from actual supplier scope 1+2 emissions. Most sustainability strategies are leaning towards primary data rather than modeled. Is there a pathway to integrate supplier-reported data, and how might the resulting values change? Could they be higher, lower, or would that vary by location? Why should we trust that your LCA values are accurate?

- How are the decarbonization strategies of oil companies considered here? Demand might be falling, but are there companies that are investing in scope 1+2 emissions reduction, and on what timelines?

- The elephant in the room is the combustion emissions from the oil. Is it possible to pull from another study to at least mention how those emissions are now and how they might change over time?

- Consider the transportation carbon intensity included in this article:
<https://doi.org/10.1016/j.trd.2020.102587>. How do your values compare?

- How can buyers of crude or refined oil integrate these values into their procurement or carbon accounting practices? How should LCA practitioners consider these values?

- The implications for policy reads more like a method than recommendations. This section could be improved to be more actionable and practical. What is the bottom line? And how can it be tied to corporate policies?

- Excellent use of visuals. The figures are extremely well done and put this paper in a top tier. I have a few comments on the figures, but they are minor considering how high quality these originals are.

- Suggestion to be consistent in naming of the life cycle phases throughout the doc and especially in the figures.

- Figures 3 and 7 are a bit confusing with the volume and carbon intensity swapping sides. Can these figures be consistent?

- Figure 4 - does everyone know what a violent plot is? It looks very compelling but the reader is not instructed how to read it.

- Figure 6 - more specifically mention transport in either the y-axes or the average line to differentiate from the production intensities

Reviewer #2 (Remarks to the Author):

Thank you for the opportunity to review “Carbon intensity of global crude oil trading and market policy implications”. The article contributes a more granular analysis that comprises a network of the global supply chain where crude blending is optimized at the scale of countries. While the contribution is valuable, the article is not acceptable in its present form.

While the estimates for emissions reductions are useful, it is unclear how the granular estimates and potential reductions compare to existing estimates. The authors develop estimates that demonstrate the variability in global “well-to-refinery-entrance” carbon emissions intensity.

For example, the article notes that “coupled with oil supply forecasts under 1.5°C scenarios up to 2050, this variability translates to additional CO₂eq savings of 1.6-5.7 Gt that could be realized solely by prioritizing low-carbon supply chain pathways without other capital-intensive mitigation measures.”

Without comparisons to existing baselines, it is unclear whether the emissions reduction potential might be over or underestimated compared to existing knowledge (rather than just applying the CI results from the analysis alone). The IEA publishes a large amount of data pertaining to the energy sector. I would recommend more direct and clear comparisons throughout the paper to elucidate potential benefits relative to present knowledge of the emissions.

Another concern is that the emissions savings are impressive, but not overly high. The authors state that present emissions equate to approximately 1.5GT/year. Between now and 2050, the CO₂eq savings of lower carbon supply chains would result in 1.6-5.7 Gt reduction, which—at its more optimistic—is about 15% per year, so 0.2GT for the highest case (which is very low compared to annual emissions). It is easy to thus question this option for decarbonizing the transportation sector. How does this option compare to others on a life cycle basis? It is not convincing with the relatively low annual emissions reductions that this path provides a substantial contribution to meeting Paris goals. The authors should clarify further.

Finally, the comparison of oil transportation options is unclear. First, are the pipeline miles described in line 270-273 a simplified representation of those in operation? Do the authors have access to a full coverage global dataset? It would also be beneficial to compare the emissions by distance (including

miles) for pipelines and shipping. I find it hard to believe that there are not substantial gaps to global pipeline coverage. The paper needs a dedicated section in the SI that better describes data coverage and data gaps, with direct note of the particular sources used.

We would like to thank the reviewers for taking the time to read our paper and provide their feedback. We have made significant updates to the main paper and the supplementary information to provide the additional discussion and comparisons that were requested, and we think these changes have strengthened the paper. We have provided detailed responses to the reviewers' comments below, with the original comments in blue and our responses in black.

Reviewer #1 (Remarks to the Author):

I think this is a well-done and interesting article that should be published after a few issues are addressed. The analysis is robust and the visuals are excellent.

1. The paper could be improved by building out what the government and corporate implications of this information should be. More specific and actionable recommendations would make the paper more meaningful. Statistics and infographics are great, but what should I do with them?

While the LCA techniques seem robust, the authors should address how the values might be different from actual supplier scope 1+2 emissions. Most sustainability strategies are leaning towards primary data rather than modeled. Is there a pathway to integrate supplier-reported data, and how might the resulting values change? Could they be higher, lower, or would that vary by location? Why should we trust that your LCA values are accurate?

How can buyers of crude or refined oil integrate these values into their procurement or carbon accounting practices? How should LCA practitioners consider these values?

The implications for policy reads more like a method than recommendations. This section could be improved to be more actionable and practical. What is the bottom line? And how can it be tied to corporate policies?

Our study is a building block toward enabling proper energy traceability, supporting governments and corporation in better understanding the impact of their decarbonization strategy from the wells to the markets. To add further emphasis to this view and in response to the feedback, we have added more detail to the “policy implications” section (lines 409-421). Specifically, we have added examples of the policy channels that could be relevant to make use of the analysis and have highlighted how the study aims to facilitate eventual convergence of model-based approaches to CI estimation and reporting practices. This convergence is particularly salient since typically, there are challenges in making like-for-like comparisons between LCA approaches and ESG reporting due differences in assumptions and system boundaries. We have indicated that in the “scope and limitations section” (lines 125-129). Additionally, lines 59-65 in the main paper introduce the potential mechanism of associating crude blends with their CI signatures that could be relevant to the policy measures needed to realize these emissions reductions.

2. How are the decarbonization strategies of oil companies considered here? Demand might be falling, but are there companies that are investing in scope 1+2 emissions reduction, and on what timelines?

The elephant in the room is the combustion emissions from the oil. Is it possible to pull from another study to at least mention how those emissions are now and how they might change over time?

We have added an explanation (lines 399-405) about how the analyzed decarbonization approach based on crude differentiation manifests in addition to and not instead of process-oriented decarbonization approaches. Furthermore, this section references a new section of the S.I. (Supplementary Note 9) to:

- i. Contextualize the emissions reduction opportunity based on the proposed market-oriented CI-based crude differentiation approach with process-oriented decarbonization approaches (e.g. scope 1+2 reduction strategies like reduced flaring, carbon capture)
- ii. Compare well-to-refinery entrance emissions with well-to-wheel emissions and overall sector emissions (including combustion)

3. Consider the transportation carbon intensity included in this article: <https://doi.org/10.1016/j.trd.2020.102587>. How do your values compare?

Building on this feedback, have added the comparison to previous aggregate quantifications of crude shipping emissions (lines 303-305), with a dedicated section added in the S.I. (Supplementary Note 8) to compare midstream emissions with relevant studies in literature. Specifically, the note compares overall midstream emissions with a separate comparison for shipping emissions due to the fact that maritime emissions exhibit wide coverage in literature. Additionally, Supplementary Note 9 also contains a comparison of the “well-to-refinery entrance” CI from our analysis with that from the IEA’s literature.

4. Excellent use of visuals. The figures are extremely well done and put this paper in a top tier. I have a few comments on the figures, but they are minor considering how high quality these originals are.

Suggestion to be consistent in naming of the life cycle phases throughout the doc and especially in the figures.

Figures 3 and 7 are a bit confusing with the volume and carbon intensity swapping sides. Can these figures be consistent?

Figure 4 - does everyone know what a violin plot is? It looks very compelling but the reader is not instructed how to read it.

Figure 6 - more specifically mention transport in either the y-axes or the average line to differentiate from the production intensities

Figure 3 has been updated with the same order of volume and CI as that of Figure 7.

The caption of Figure 4 has been updated with more detail about the violin plot and how it encodes the distribution of C.I. values for the listed crude blends.

“Transport C.I.” has been specifically added to the description of the y-axes of Figure 6.

Reviewer #2 (Remarks to the Author):

Thank you for the opportunity to review “Carbon intensity of global crude oil trading and market policy implications”. The article contributes a more granular analysis that comprises a network of the global supply chain where crude blending is optimized at the scale of countries. While the contribution is valuable, the article is not acceptable in its present form.

1. While the estimates for emissions reductions are useful, it is unclear how the granular estimates and potential reductions compare to existing estimates. The authors develop estimates that demonstrate the variability in global “well-to-refinery-entrance” carbon emissions intensity.

For example, the article notes that “coupled with oil supply forecasts under 1.5°C scenarios up to 2050, this variability translates to additional CO₂eq savings of 1.6-5.7 Gt that could be realized solely by prioritizing low-carbon supply chain pathways without other capital-intensive mitigation measures.”

Without comparisons to existing baselines, it is unclear whether the emissions reduction potential might be over or underestimated compared to existing knowledge (rather than just applying the CI results from the analysis alone). The IEA publishes a large amount of data pertaining to the energy sector. I would recommend more direct and clear comparisons throughout the paper to elucidate potential benefits relative to present knowledge of the emissions.

To address this feedback around providing relevant baselines, we’ve added relevant content to the main paper and have created dedicated sections in the SI for additional details.

First, the total annual “well-to-refinery-entrance” emissions provided by the IEA has been added to the “main” section of the main paper, line 38. This value was originally obtained from “Global carbon intensity of crude oil production” Masnadi et al.; since we use the field-level C.I. values from this study, we have aimed to address the feedback using the aforementioned independent reference from the IEA. Similarly, IEA’s estimate for well-to-refinery entrance CI has been provided for direct comparison in the “Implications for decarbonization policy” section of the main paper (lines 372-373).

Second, to facilitate a direct comparison of the midstream C.I. values with relevant literature, we’ve added a dedicated section – Supplementary Note 8 and have referenced that in the main paper (lines 217, 303-305.) Specifically, the note compares

- i. Overall midstream emissions with the IEA’s literature as a reference
- ii. Shipping emissions with multiple relevant studies, due to the fact that maritime emissions have exhibited wide coverage in literature

Another concern is that the emissions savings are impressive, but not overly high. The authors state that present emissions equate to approximately 1.5GT/year. Between now and 2050, the CO₂eq savings of lower carbon supply chains would result in 1.6-5.7 Gt reduction, which—at its more optimistic—is about 15% per year, so 0.2GT for the highest case (which is very low compared to annual emissions). It is easy to thus question this option for decarbonizing the transportation sector. How does this option compare to others on a life cycle basis? It is not convincing with the relatively low annual emissions reductions that this path provides a substantial contribution to meeting Paris goals. The authors should clarify further.

The emissions reduction opportunities that our study highlights (i.e. market-oriented informed by CI-based crude differentiation are different from A) the evolving energy mix (e.g. switch to renewables) and B) process-oriented decarbonization within the supply chain (e.g. reduced flaring). Moreover, we note that these opportunities are complementary and can be realized in addition to, rather than instead of each other.

To that end, we have added a dedicated section in the S.I. (Supplementary Note 8) to:

- i. Contextualize the emissions reduction opportunity based on the proposed market-oriented CI-based crude differentiation approach with process-oriented decarbonization approaches (e.g. scope 1+2 reduction strategies like reduced flaring, carbon capture)
- ii. Compare well-to-refinery entrance emissions with well-to-wheel emissions and overall sector emissions (including combustion)

We have also added explanation in the main paper (lines 399-405) about how the analyzed decarbonization approach based on crude differentiation manifests in addition to and not instead of process-oriented decarbonization approaches referencing the above supplementary note.

With the details added to the aforementioned supplementary note in context, we note that market-based solutions which would incentivize CI-based crude differentiation offer a reduction potential comparable to that of the process-oriented approaches at a significantly lower capital intensity (Supplementary Note 8, S.I. lines 438-450).

2. Finally, the comparison of oil transportation options is unclear. First, are the pipeline miles described in line 270-273 a simplified representation of those in operation? Do the authors have access to a full coverage global dataset? It would also be beneficial to compare the emissions by distance (including miles) for pipelines and shipping. I find it hard to believe that there are not substantial gaps to global pipeline coverage. The paper needs a dedicated section in the SI that better describes data coverage and data gaps, with direct note of the particular sources used.

The pipeline dataset underpinning the analyses is a commercially-sourced dataset from GlobalData (<https://www.globaldata.com/marketplace/oil-and-gas/midstream-pipelines/>) that provides a high-fidelity, global coverage. We have added relevant details on dataset coverage and limitations under the respective sections of Supplementary Note 1. In addition, we have also added examples of how the pipeline dataset is used to construct the network in Supplementary Figure 3.

We find that while the commercial dataset is comprehensive, the network requires the construction of suitable edges to render meaningful connectivity as detailed in Supplementary Note 3. This is in direct connection to the observation of potential gaps in the pipeline dataset, specifically in regards to auxiliary pipelines or intra-field pipelines. We have further quantified the extent to which “constructed” edges in the network are used in comparison to pipelines in terms of total barrel-miles (Supplementary Note 1, S.I. lines 86-89) – wherein we note that along non-shipping edges, the split of total barrel-miles in the network is 92% along pipeline edges (i.e. edges mapped to pipeline segments from the dataset) and 8% are along these “constructed” edges.

Lastly, to add more context to the estimation of transportation emissions, we have added more information on the scale of the different modes and emissions per unit distance in

the main paper lines 306-310.

REVIEWER COMMENTS

Reviewer #2 (Remarks to the Author):

The work is very nicely done and represents an important contribution; however, two important items remain unaddressed.

I am generally pleased with the revisions with one exception: the extent of the pipeline coverage. It appears that the authors response and analysis concentrate more on transmission rather than gathering pipelines. The former tends to have much more robust coverage than the latter. The coverage of the latter is quite poor. That said, presumably gathering infrastructure (i.e., pipelines connecting wells to main transmission lines and processing facilities) will also have an impact on CI. The authors should address this more directly; e.g., could results underestimate CI since highly resolved gathering infrastructure is not available across all regions? Additionally, it should be clear that the pipeline coverage pertains to transmission infrastructure. To demonstrate a comprehensive gathering infrastructure dataset would be quite a substantial effort, requiring comprehensive well-level data.

Second, the pipeline information prompted me to think more upstream and look more carefully for the methodology and data used to characterize methane emissions. Too frequently, the functional unit is not employed correctly while not correctly allocating; for example, the numerator represents emissions of BOTH oil and gas. The authors should include a short section in the SI that specifically addresses how methane emissions from the upstream supply chain were quantified and the data sources employed.

Reviewer #3 (Remarks to the Author):

The manuscript is well-written. However, there are some comments to improve the manuscript quality

1. Please use a moderate title that represents the topic discussed (refer to comment no 4)
2. Line 20: The word “evolving” is exaggerated. I think using “energy mix transition” is better for the sentence
3. Line 22: The sentence is not showing a background study. Please revise the background statement of the carbon intensity of global crude oil trading or delete the sentence

4. Line 27: The word “global” is exaggerated since the data presented is limited to less than 50 countries. Instead of using “global” in the abstract, please mention the number of studied countries.
5. Line 35: Please add a reference from IPCC or Paris Agreement statement for global warming projection
6. Line 39: Is the number from the current study or reference? If it comes from another study, please add the reference for these sentences
7. Line 41: Please support the sentence with reference
8. Line 46: Since the topic of this manuscript is global energy mix, please add literature with other carbon policy in the different continent.
9. Line 72: Please improve the background statement and explain why this study is matter? What is the novelty of the study compared with previous study
10. Line 78: Please move the detailed scope and resolution of LCA to the methodology and replace it with the headed result and discussion
11. Please state the noteworthy results in the main section
12. Line 100: condensate is an unseparate product of crude oil. Please write detailed method how the author excluding condensate from the estimation and analysis
13. Line 103: what is time average LCA? Is it static LCA (time is not included as analysis variable)?
14. Line 108: How many percentage is transportation covered in this study if the the barges, truck and railway is excluded? Please add a sentence which explain the coverage percentage of studied transportation compared to global transport used.
15. Fig 1: please increase the font size in the figure
16. Fig 2: please increase the font size in the figure
17. Line 175: is there any particular reason why south asia, east asia or south-east is asia is not included in the global crude oil blends carbon intensity?
18. Line 268: is the sentence based on the data study or literature review? Please put a citation if from literature review.
19. Line 333: How many percentage the countries that process >1 million barrles of crude oil per day compared to the total country existed? Please add explanation to measure whether it is apporiate representative of global scope or not.
20. Line 347: why greater pipeline has higher transportation CI? Please expand the discussion that compare the pipeline miles with the other crude oil transportation?
21. Line 370: Please put the main decarbonization policy/strategie in the beginning of sub-section and followed by the specific discussion

22. How do you explain the work is significance to the field and related field? Please add novelty statement on how difference of current work compared to previous work

23. Line 436: Please add the detailed data collection in the method

24. Line 436: Please add the boundary study in the methodology

We thank the reviewers for taking the time to read our paper and provide their feedback. The reviewer feedback has helped us make this paper more informative and understandable. Of particular note is the use of an updated version of OPGEE to help address the questions about emissions associated with field-level gathering pipelines. Since this required re-running several components of the analysis to propagate the effects of this update, many numerical results in the paper have changed as a result. We have edited the text and figures and have provided detailed responses to the reviewers' comments below (original comments are in blue and our responses in black).

Reviewer #2 (Remarks to the Author):

The work is very nicely done and represents an important contribution; however, two important items remain unaddressed.

I am generally pleased with the revisions with one exception: the extent of the pipeline coverage. It appears that the authors response and analysis concentrate more on transmission rather than gathering pipelines. The former tends to have much more robust coverage than the latter. The coverage of the latter is quite poor. That said, presumably gathering infrastructure (i.e., pipelines connecting wells to main transmission lines and processing facilities) will also have an impact on CI. The authors should address this more directly; e.g., could results underestimate CI since highly resolved gathering infrastructure is not available across all regions? Additionally, it should be clear that the pipeline coverage pertains to transmission infrastructure. To demonstrate a comprehensive gathering infrastructure dataset would be quite a substantial effort, requiring comprehensive well-level data.

The differentiation between transmission and gathering pipelines manifests in two main ways:

First, the enhancements to the OPGEE model we have made to estimate upstream carbon intensities at the field level enable a clearer distinction between emissions from the gathering and transmission systems. Given the challenges associated with data pertaining to gathering pipelines, we use assumptions around gas gathering / gas flaring to account for emissions from the gathering systems at the field level. This has been further expanded on in the S.I., Supplementary Note 10.

Second, the availability of data and the assumption to represent supply chain entities as point objects rather than spatial objects. The three main transportation edge types represented in the supply chain network are pipeline, shipping and other. The "other" edges are heuristically constructed as described in the S.I. and can conceptually interpreted as the intra-field pipeline connections and approximations to substitute for missing pipeline data. Thus, the emissions associated with intra-field connectivity i.e. well to processing / transmission junctions is approximated using these "constructed edges". Furthermore, based on the approximation, we observe that emissions associated with these auxiliary pipelines are 8% of those associated with main pipelines, i.e. significantly smaller. To add more explanation around this, we have added and augmented the following points in the text:

- Main paper, lines 274-280: intuition behind the "other" edges, comparison with pipeline edges and reference to the S.I.

- Main paper, lines 448-457: explanation about the “other” edges in the network construction process
- Main paper, lines 468-475: assumption to represent supply chain entities as point objects
- S.I., Supplementary Note 1 / Supply chain infrastructure: coverage of pipeline v/s other edges

Second, the pipeline information prompted me to think more upstream and look more carefully for the methodology and data used to characterize methane emissions. Too frequently, the functional unit is not employed correctly while not correctly allocating; for example, the numerator represents emissions of BOTH oil and gas. The authors should include a short section in the SI that specifically addresses how methane emissions from the upstream supply chain were quantified and the data sources employed.

We have expanded our discussion (main paper lines 419-433) of how we account for the coproduction of oil and condensate. We defer to the method used by Masnadi et al. (2018), which categorizes oil and gas wells using a gas-to-oil ratio threshold of 10,000 scf/well. Wells below this threshold are oil wells and are included in our analysis; wells above this threshold are natural gas wells and are not included in our analysis. For wells that are categorized as oil wells, we use energy-based coproduct allocation to split emissions between oil and condensate products. This approach is consistent with ISO 14044 recommendation for coproduct management: we use gas-to-oil ratio as way to partition oil and gas systems to avoid allocation between oil and gas products, and then we use a physical basis (energy) to allocate between the remaining coproducts (oil and condensate).

Reviewer #3 (Remarks to the Author):

The manuscript is well-written. However, there are some comments to improve the manuscript quality

1. Please use a moderate title that represents the topic discussed (refer to comment no 4)

Please refer to the response for comment number 4

2. Line 20: The word “evolving” is exaggerated. I think using “energy mix transition” is better for the sentence

We have incorporated the proposed alternative framing.

3. Line 22: The sentence is not showing a background study. Please revise the background statement of the carbon intensity of global crude oil trading or delete the sentence

Due to the manuscript requirement of not including citations in the abstract, we would like to mention the text in the main section that corresponds to this sentence: Lines 47-60 refer to

the current regulatory efforts and the limitations of current estimates on the basis of resolution and supply chain coverage. We have included citations for the appropriate references used in this section.

4. Line 27: The word “global” is exaggerated since the data presented is limited to less than 50 countries. Instead of using “global” in the abstract, please mention the number of studied countries.

Our assessment exhibits global coverage both geographically and in terms of crude production volumes.

- Production volume coverage:
As indicated in the main paper (lines 422-429), the assessment includes crude supply of ~95% of the combined global crude oil and condensate production.
- Geographical coverage:
 - The assessment includes 64 producer countries, 83 consumer countries with coverage across all populated continents (North America, South America, Africa, Europe, Asia and Oceania).
 - In comparison with Masnadi et al. (Global carbon intensity of crude oil production), which is the source for upstream field-level CI estimates used by our study, the following countries are excluded from the source side of the field to refinery tracking:

South Sudan, Niger, Netherlands, Afghanistan, Guatemala, Tajikistan, Pakistan, Cote d'Ivoire, Kyrgyzstan, Spain, Suriname, Jordan, Israel, Japan, Georgia, Bolivia, Greece, Morocco, Belize

This selection is done due to the absence of crude blends originating from the corresponding countries in the market trades dataset. This exclusion is salient given the assumption of the blend estimation algorithm which estimates blend formation from oil fields originating from the same country:

These exclusions represent less than 0.5% of global field-level production volume. Note that, from these countries, all those which contain refining capacity are included in the destination side of the field to refinery tracking.

We have added a section in the S.I. titled “Geographical coverage of the supply chain” to explicitly include the above information.

Lastly, we would also like to point to S.I. Tables 6 and 9, which include upstream and midstream C.I. estimates for blends with volume >100,000 barrels/day. This set represents 90.2% of the total blend-level production volume included in the assessment and hence provides the key outputs from the study, which span globally, in a readable format.

5. Line 35: Please add a reference from IPCC or Paris Agreement statement for global warming projection

We have added a reference to the IPCC guidelines for policy makers.

6. Line 39: Is the number from the current study or reference? If it comes from another study, please add the reference for these sentences

The number is from IEA's assessment: <https://www.iea.org/data-and-statistics/charts/spectrum-of-the-well-to-tank-emissions-intensity-of-global-oil-production-2018>; the reference for which has been included.

7. Line 41: Please support the sentence with reference

We have added two references (Sahebi and Beiranvand) that point to the complexity of the supply chain and have included a reference to the supply chain infrastructure datasets that are used in the study.

8. Line 46: Since the topic of this manuscript is global energy mix, please add literature with other carbon policy in the different continent.

In addition to the policies of CARB (California Air Resources Board) and the EU's (European Union) FQD (Fuel Quality Directive), we have added CORSIA (Carbon Offsetting and Reduction Scheme for International Aviation), a global policy for jet fuels to give a more global context.

9. Line 72: Please improve the background statement and explain why this study is matter? What is the novelty of the study compared with previous study

We have updated the content in lines 75-82 to emphasize the novelty statement for our study, which mainly constitutes A) high-resolution assessment (granular supply chain pathways), B) end-to-end coverage from fields to refineries C) bottom-up engineering-based methods and D) global coverage ~95% crude supply across all major populated continents E) CI estimates for varying levels of aggregation, most noteworthy from a policy standpoint being those at the marketed crude-blend level.

10. Line 78: Please move the detailed scope and resolution of LCA to the methodology and replace it with the headed result and discussion

Addressed together with comment number 11; please refer below

11. Please state the noteworthy results in the main section

Based on this comment and the previous, we have suitably reorganized the sections. Given the submission guidelines, the section “Main” serves as an introduction to the key research question with details around background information. We have then kept the “Scope and resolution of the LCA” section brief to only include details that are necessary to contextualize the results that follow. The remaining parts that were contained in this section have been moved to the “Methods” section at the end of the main paper. Following the “Scope and resolution of the LCA” section, we then have the key results-oriented sections – “Carbon intensity of marketed global crude oil blends”, “Crude transportation CI from producer to consumer countries” and “Net carbon footprint attributed to consumer countries”.

12. Line 100: condensate is an unseparate product of crude oil. Please write detailed method how the author excluding condensate from the estimation and analysis

We have expanded our discussion of how we account for coproduction of oil and condensate. We defer to the method used by Masnadi et al. (2018), which categorizes oil and gas wells using a gas-to-oil ratio threshold of 10,000 scf/well. Wells below this threshold are oil wells and are included in our analysis; wells above this threshold are natural gas wells and are not included in our analysis. For wells that are categorized as oil wells, we use energy-based coproduct allocation to split emissions between oil and condensate products. This approach is consistent with ISO 14044 recommendation for coproduct management: we use gas-to-oil ratio as way to partition oil and gas systems to avoid allocation between oil and gas products, and then we use a physical basis (energy) to allocate between the remaining coproducts (oil and condensate).

13. Line 103: what is time average LCA? Is it static LCA (time is not included as analysis variable)?

We have replaced the phrase “time averaged” with “static, annualized” to better reflect the LCA. Specifically, we take the per day barrel throughput across the supply chain and annualize it (multiply by 365 days/year) to report total emissions on a per year basis.

14. Line 108: How many percentage is transportation covered in this study if the the barges, truck and railway is excluded? Please add a sentence which explain the coverage percentage of studied transportation compared to global transport used.

While globally the percentage share of rail, barges and trucking is not known broadly, to the best of our knowledge, we have added references (lines 459-466) from governmental reporting from the U.S. to act as a directional indicator (rail, barges and trucks cover ~10% crude transport based on refinery intake receipts). Additionally, to add more context around the percentage share in the U.S., we note the presence of non-oceanic water bodies and a mature crude rail transportation system.

15. Fig 1: please increase the font size in the figure

We have increased the font size, particularly of in-figure annotations and legend items to improve readability.

16. Fig 2: please increase the font size in the figure

We have increased the font size, particularly of in-figure annotations and legend items to improve readability.

17. Line 175: is there any particular reason why south asia, east asia or south-east is asia is not included in the global crude oil blends carbon intensity?

Countries from South Asia, East Asia, South-east Asia are included in the assessment:

- China, Thailand, Vietnam included in S.I. Table 6 and Table 9
- All major consumer countries from the region (e.g. India, China, Japan, South Korea etc.) included in Fig 5, 6 and 7 of Main Paper
- Note: Since this region doesn't include a produced blend in the top 20 global blends by volume, it is excluded in Fig 4

18. Line 268: is the sentence based on the data study or literature review? Please put a citation if from literature review.

This sentence is based on the midstream supply chain infrastructure dataset provided by GlobalData. We have added the appropriate references to indicate the same.

19. Line 333: How many percentage the countries that process >1 million barrels of crude oil per day compared to the total country existed? Please add explanation to measure whether it is appropriate representative of global scope or not.

The analysis includes all countries with refining capacity based on the downstream dataset provided by Wood Mackenzie. Based on which, in Figure 7, the geographical choropleth map includes all the countries being assessed (without the filter for >1 Mbbbl/d); countries with refining volume >1 Mbbbl/d are shown in the adjoining bar graph with the intention to show additional information on how the major refining countries compare. This is done to supplement the comparison shown in the map. These (countries with refining volume >1 Mbbbl/d) represent 78% of the total refining volume included in the study. We have added more explanation in Lines 303-310 to provide this context around Figure 7.

20. Line 347: why greater pipeline has higher transportation CI? Please expand the discussion that compare the pipeline miles with the other crude oil transportation?

We have added more explanation for the dependence of pipeline CI on distance in lines 321-325. Additionally, given this insight stems from the fluid-mechanics based approach

COPTTEM used in the study, we have also added an additional note in the S.I with mentions to the appropriate references and assumptions. Overall pipeline barrel-miles are compared to shipping miles in lines 281-285.

21. Line 370: Please put the main decarbonization policy/strategie in the beginning of subsection and followed by the specific discussion

We have reorganized the “Policy Implications” section suitably and have put the main policy-oriented discussion at the beginning in lines 353-370.

22. How do you explain the work is significance to the field and related field? Please add novelty statement on how difference of current work compared to previous work

We have added more context around significance and novelty of our study; please refer to comment number 9.

23. Line 436: Please add the detailed data collection in the method

We have added the key data sources in lines 409-417 and have referenced the S.I (Supplementary Note 1) for details on the data sources, key attributes, coverage and limitations.

24. Line 436: Please add the boundary study in the methodology

Our response to comment number 12 addresses this comment.

REVIEWER COMMENTS

Reviewer #2 (Remarks to the Author):

The reviews were generally nicely done. I see 3 key items that remain that can be addressed in minor revisions.

1. The revisions in the main text do not differentiate between gathering and transmission. Key data gaps should be clearer with specific reference (gathering and boosting infrastructure is well known to contribute to emissions, so the length of these pipelines matter). It otherwise appears like a purposeful omission.
2. OPGEE has improved, but no model is gospel. The choice of methane emissions estimates should be better described, including limitations and other options (i.e., why OPGEE was chosen but also how it is limited).
3. A short section in the SI describing specific data gaps and future research would strengthen the contribution.

Reviewer #3 (Remarks to the Author):

The comments and reviews have been responded to clearly. I suggest we accept the manuscript

We would like to thank the reviewers for their additional suggestions for our paper. We have revised the manuscript and supplementary information to provide additional context and references to address the reviewers' concerns and provided a description of these changes below. The reviewer comments are in green, with our responses in black.

1. The revisions in the main text do not differentiate between gathering and transmission. Key data gaps should be clearer with specific reference (gathering and boosting infrastructure is well known to contribute to emissions, so the length of these pipelines matter). It otherwise appears like a purposeful omission.

Response: We agree that emissions originating from natural gas gathering and boosting infrastructure (i.e., pipelines) can play a key role in estimating oil and gas production emissions. These pipelines are most often operated via compressor stations at a pressure (100-2,000 kPa) that is lower than transmission pipelines but higher than distribution pipelines. Methane emissions can be found in three main sources, including underground pipeline leaking, above-ground auxiliary equipment leaking, and intentional venting (e.g., maintenance blowdown)^{1,2}. To date, few studies have characterized emissions from such infrastructure using the top-down approach because the scale and complexity of gathering pipeline network and the fact that many are underground and in difficult-to-access locations.

U.S. EPA³ reported that in 2021 emissions come from gathering and boosting infrastructure are almost on par with those from oil and gas production emissions on an absolute basis (both around 90 million metric tons CO₂e). A nation-wide gathering pipeline emissions factor of 0.19 Mg year⁻¹ km⁻¹ considering 710,000 km gathering pipeline was concluded by U.S. EPA⁴. Several other studies⁵⁻⁷ focusing on ground-based surveys showed that gathering pipeline emissions are 0.75 Mg year⁻¹ km⁻¹ at the most. Yu et al.⁸ conducted aerial-based campaigns in the Permian basin and reported that emissions factor estimates for gathering pipelines can range from 2.7 (+1.9/-1.8) Mg year⁻¹ km⁻¹ for Fall 2021 to 10.0 (+6.4/-6.2) Mg year⁻¹ km⁻¹ for Fall 2019. The discrepancy in estimates between the ground- and aerial-based studies may be due to area covered and sampling size. Therefore, the distance of gathering pipelines can be a crucial factor in estimating such fugitive emissions.

However, it is noteworthy that the vast majority of gathering and boosting emissions are associated with natural gas production and processing. In this study, we focus on oil producing fields with a cutoff gas-to-oil ratio at 10,000 scf/bbl, meaning that any fields producing more gas than this cutoff ratio are excluded. Therefore, the contribution of gathering and boosting emissions to the emissions factor for crude oil is much less than that for natural gas. In addition, information about global gathering and boosting pipelines is not available for estimating the associated emissions on a bottom-up basis. Nonetheless, OPGEE does have a component-level fugitive model that estimates methane emissions from storage tank and equipment leakage based on a comprehensive literature review of component-level activity and emissions measurements⁹. This may help cover part of the aforementioned gathering and boosting emissions but is less ideal than having bottom-up estimates using gathering and boosting pipeline infrastructure information

(e.g., distance) and region-specific emissions factors. To address the review's comment and facilitate the readers, we have made the following changes to the main text:

Line 433-440, Methods:

“Emissions from natural gas gathering and boosting infrastructure can be a significant contribution to oil and gas production emissions. However, in this study, data are not available for estimating such emissions on a bottom-up basis. OPGEE has a component-level fugitive model that estimates methane emissions from storage tank and equipment leakage based on a comprehensive literature review of component-level activity and emissions measurements. Nonetheless, more data about gathering and boosting infrastructure and fugitive emissions measurements could help improve the accuracy of OPGEE and hence the results of this study (refer to S.I. more details).”

S.I., Supplementary Note 11 Data Gaps and Future Research:

“Natural gas gathering and boosting infrastructure (i.e., pipelines) can play a key role in oil and gas production emissions. These pipelines are most often operated via compressor stations at pressures of 100–2,000 kPa, which is lower than transmission pipelines but higher than distribution pipelines. Methane emissions can originate from three main sources, including underground pipeline leaking, above-ground auxiliary equipment leaking, and intentional venting (e.g., maintenance blowdown)^{1,2}. To date, few studies have characterized emissions from such infrastructure using a top-down approach because the scale and complexity of gathering pipeline networks and the fact that many are underground and in difficult-to-access locations.

U.S. EPA³ reported that in 2021, emissions from gathering and boosting infrastructure were almost on par with those from oil and gas production emissions on an absolute basis (both around 90 million metric tons CO₂e). A nation-wide gathering pipeline emissions factor of 0.19 Mg year⁻¹ km⁻¹ considering 710,000 km gathering pipeline was estimated by U.S. EPA⁴. Several other studies⁵⁻⁷ focusing on ground-based surveys showed that gathering pipeline emissions are 0.75 Mg year⁻¹ km⁻¹ at the most. Yu et al.⁸ conducted aerial-based campaigns in the Permian basin and reported that emissions factor estimates for gathering pipelines can range from 2.7 (+1.9/–1.8) Mg year⁻¹ km⁻¹ for Fall 2021 to 10.0 (+6.4/–6.2) Mg year⁻¹ km⁻¹ for Fall 2019. The discrepancy in estimates between the ground- and aerial-based studies may be due to area covered and sampling size. Therefore, the distance of gathering pipelines can be a crucial factor in estimating such fugitive emissions.

However, it is noteworthy that the vast majority of gathering and boosting emissions are associated with natural gas production and processing. In this study, we focus on oil producing fields with a cutoff gas-to-oil ratio at 10,000 scf/bbl, meaning that any fields producing more gas than this cutoff ratio are excluded. Therefore, the contribution of gathering and boosting emissions to the emissions factor for crude oil is much less than that for natural gas. In addition, information about global gathering and boosting pipelines is not available for estimating the associated emissions on a bottom-up basis. Nonetheless, OPGEE v3.0 does have a component-level fugitive emissions model that estimates methane emissions from storage tank and equipment leakage based on a comprehensive literature review of component-level activity and emissions

measurements⁹. This may help cover part of the aforementioned gathering and boosting emissions but is less ideal than having bottom-up estimates using gathering and boosting pipeline infrastructure information (e.g., distance) and region-specific emissions factors.”

2. OPGEE has improved, but no model is gospel. The choice of methane emissions estimates should be better described, including limitations and other options (i.e., why OPGEE was chosen but also how it is limited).

Response: The choice of methane emissions estimates has been described in “Supplementary Note 10: Relevant details on estimation of field-level upstream emissions” in the S.I. This note explains how OPGEE estimates fugitive emissions based on a comprehensive publicly available database of component-level activity and emissions measurements, encompassing six studies and approximately 3,200 measurements. We have included the following section in the S.I. to address the reviewer’s concern about limitations and other options.

S.I., Supplementary Note 11 Data Gaps and Future Research:

“.....This may help cover part of the aforementioned gathering and boosting emissions but is less ideal than having bottom-up estimates using gathering and boosting pipeline infrastructure information (e.g., distance) and region-specific emissions factors.”

“The fugitive emissions model OPGEE v3.0 employs is constructed upon the most comprehensive public datasets available to date, encompassing component-level equipment counts and emissions⁹. However, these datasets are based solely on United States data. This specificity introduces certain limitations, given the differences between the upstream oil and gas operations in the United States and other regions worldwide.

In particular, the extrapolation of the US-centric data to a global scale may generate errors in estimating component-level loss rates, especially considering the variability and heterogeneity of operational practices, technologies, and regulatory standards across the globe. Such discrepancies highlight the value of integrating region-specific datasets and developing a globally representative and accurate model.

In addition, a secondary constraint within the model’s design relates to the scope of emissions it can accurately predict. As depicted in Supplementary Figure 35 by Rutherford et al.⁹, the model’s predicted maximum well-site CH₄ emissions cap at 100 kg/h. However, evidence provided by Chen et al.¹⁰ via a comprehensive aerial survey delineates a broader range for well-site CH₄ emissions, extending from 10 kg/h to as high as 10,000 kg/h.

Analysis of the probability density distribution indicates the peak of CH₄ emissions to be approximately around 100 kg/h. However, when evaluating the cumulative density distribution, emissions surpassing 100 kg/h account for a substantial 75% of total emissions. This substantial underestimation of emissions intensity highlights the necessity for model adjustments to encompass higher-end emissions values. Overall, these identified limitations point towards the

potential for developing more comprehensive and geographically representative models to accurately estimate emissions in the global upstream oil and gas sector.”

We have also included the following section in “Supplementary Note 10 Relevant details on estimation of field-level upstream emissions” in the S.I. to demonstrate why OPGEE is selected for estimating upstream emissions in this study.

Supplementary Note 10 Relevant details on estimation of field-level upstream emissions

“Supplementary Table 10 compares various emission estimation methods used in the upstream oil and gas sector. Among these methods, OPGEE showcases significant advantages over its counterparts.

Comparing OPGEE with other bottom-up models, two major strengths are immediately discernible. Firstly, OPGEE is an open-source model that promotes transparency and allows for public validation and verification, a feature often lacking in other models. Secondly, the breadth of production methods that OPGEE can model surpasses other bottom-up approaches. It encompasses a diverse range of methods, such as gas lifting and flooding, and includes a broader selection of processing units.

When compared with the U.S. Environmental Protection Agency’s Greenhouse Gas Reporting Program (GHGRP), OPGEE’s capability extends beyond the confines of US fields, rendering it a more versatile and globally applicable model.

In relation to top-down methodologies, like aerial surveys and satellite monitoring, OPGEE has the unique advantage of computing CO₂ equivalents, which include both CO₂ and CH₄ emissions. Existing top-down techniques are predominantly limited to measuring CH₄ emissions. Furthermore, the relative error in these top-down approaches under a single-blind test can be extensive, ranging from -100% to 100%¹¹, demonstrating the superior precision of OPGEE.

Compared to commercial entities like Project Canary, which require users to install sensors on equipment and conduct measurements, OPGEE represents a more cost-effective and efficient solution. Therefore, for comprehensive, accessible, and accurate estimation of greenhouse gas emissions in the oil and gas sector, OPGEE emerges as a distinctly powerful tool.

Supplementary Table 10: Comparative Review of Emission Estimation Methods in the Upstream Oil and Gas Sector

Method Name	Method Type	Emissions Type	System Boundary	Modeling Range	Open-source
OPGEE ¹²	Bottom-up	CO ₂ e	Well-to-refinery	Global	Yes
FUNNELGHG-COO ¹³	Bottom-up	CO ₂ e	Well-to-tank	Conventional oil	No
FUNNELGHG-OS ¹³	Bottom-up	CO ₂ e	Well-to-tank	Oil sands	No

GHGRP ^{3,14}	Direct Report	CO _{2e}	Well-to-refinery	U.S.	Yes
Aerial Survey ¹⁰	Top-down	CH ₄	N.A.	Global	No
Carbon Mapper Satellite ¹⁵	Top-down	CH ₄	N.A.	Global	Yes
Project Canary ¹⁶	Direct Measurement	CH ₄	N.A.	Global	No

”

3. A short section in the SI describing specific data gaps and future research would strengthen the contribution.

Response: This is now included in Supplementary Note 11 “Data Gaps and Future Research” in the S.I., as detailed in the responses to the preceding comments.

References

1. Marchese, A. J. *et al.* Methane Emissions from United States Natural Gas Gathering and Processing. *Environ. Sci. Technol.* (2015) doi:10.1021/acs.est.5b02275.
2. Zimmerle, D. *et al.* Methane Emissions from Gathering Compressor Stations in the U.S. *Environ. Sci. Technol.* (2020) doi:10.1021/acs.est.0c00516.
3. U.S. EPA. GHGRP Petroleum and Natural Gas Systems. <https://www.epa.gov/ghgreporting/ghgrp-petroleum-and-natural-gas-systems#subsector> (2022).
4. U.S. EPA. Annex 3.6: Methodology for Estimating CH₄, CO₂, and N₂O Emissions from Natural Gas Systems. https://www.epa.gov/sites/default/files/2017-02/documents/3._6_natural_gas_systems_annex_2017-2-10_.pdf (2022).
5. Zimmerle, D. J. *et al.* Gathering pipeline methane emissions in Fayetteville shale pipelines and scoping guidelines for future pipeline measurement campaigns. *Elementa* (2017) doi:10.1525/elementa.258.
6. Li, H. Z., Mundia-Howe, M., Reeder, M. D. & Pekney, N. J. Gathering pipeline methane emissions in utica shale using an unmanned aerial vehicle and ground-based mobile sampling. *Atmosphere (Basel)*. (2020) doi:10.3390/atmos11070716.
7. Li, H. Z., Mundia-Howe, M., Reeder, M. D. & Pekney, N. J. Constraining natural gas pipeline emissions in San Juan Basin using mobile sampling. *Sci. Total Environ.* (2020) doi:10.1016/j.scitotenv.2020.142490.
8. Yu, J. *et al.* Methane Emissions from Natural Gas Gathering Pipelines in the Permian Basin. *Environ. Sci. Technol. Lett.* (2022) doi:10.1021/acs.estlett.2c00380.
9. Rutherford, J. S. *et al.* Closing the methane gap in US oil and natural gas production emissions inventories. *Nat. Commun.* (2021) doi:10.1038/s41467-021-25017-4.
10. Chen, Y. *et al.* Quantifying Regional Methane Emissions in the New Mexico Permian Basin with a Comprehensive Aerial Survey. *Environ. Sci. Technol.* (2022) doi:10.1021/acs.est.1c06458.
11. Sherwin, E. D. *et al.* Single-blind validation of space-based point-source detection and quantification of onshore methane emissions. *Sci. Reports 2023 131* (2023).

12. Masnadi, M. S. *et al.* Global carbon intensity of crude oil production. *Science* (80-.). **361**, 851–853 (2018).
13. Nimana, B., Canter, C. & Kumar, A. Energy consumption and greenhouse gas emissions in upgrading and refining of Canada’s oil sands products. *Energy* **83**, 65–79 (2015).
14. Statistics, C. *Greenhouse gas reporting program. GHGRP 2016: Reported Data* (2018).
15. Cusworth, D. H. *et al.* Strong methane point sources contribute a disproportionate fraction of total emissions across multiple basins in the United States. *Proc. Natl. Acad. Sci. U. S. A.* (2022) doi:10.1073/pnas.2202338119.
16. Project Canary. Advanced continuous emissions monitoring (cem) guide. <https://www.projectcanary.com/>.